# A new molecular tool for detection of the highly invasive gecko, *Hemidactylus frenatus*

**Arati Agarwal[1], Melissa L. Thomas[2], Yvette Hitchen[3], Paul Doughty[4], Simon J. McKirdy[2], Brendan C. Rodoni[1,5], Mark J. Blacket[1]\***

1 Agriculture Victoria Research, AgriBio - Centre for AgriBioscience, Bundoora, Victoria, Australia, 2 Harry Butler Institute, Murdoch University, Perth, Australia, 3 Helix Molecular Solutions, Leederville, Australia, 4 Western Australia Museum, Terrestrial Zoology, Welshpool, Australia, 5 School of Applied Systems Biology, La Trobe University, Bundoora, Victoria, Australia

\* mark.blacket@agriculture.vic.gov.au

## Abstract

The Common House Gecko, *Hemidactylus frenatus*, also known as the Asian House Gecko (AHG), is the most significant invasive gecko globally. Detecting this species can be challenging because it closely resembles other geckos, and is often not directly observed, being cryptic and nocturnal. Traces such as scats, however, are more readily observed than the animal itself. Here, we developed and tested a new diagnostic mitochondrial ND2 LAMP (Loop-mediated isothermal amplification) assay to detect and distinguish AHG from other geckos. Testing DNA from twenty-five non-target gecko and skink species present in Western Australia demonstrated the species-specificity of the assay. This new molecular assay showed amplification in under 15 minutes from AHG DNA. Intraspecific variation did not adversely affect the LAMP assay, with all AHG tissue samples successfully amplifying. This included samples from the Cocos (Keeling) Islands territory of Australia which were >6% genetically divergent from mainland Australian samples, representing a genetic group that was previously unknown, referred to here as AHG "clade E". The assay was found to be highly sensitive, capable of amplifying AHG DNA at very low levels, down to 0.0001 ng/μL of AHG DNA, within 25 minutes. The new LAMP assay has been fully optimised for in-field use, including development and testing of a non-destructive DNA extraction method for in-field extractions from both AHG tissue and scat samples, as well as a gBlock gene fragment for use as a synthetic positive control. The in-field protocols were tested on 100 field collected scats, from multiple lizard species in Western Australia, demonstrating the AHG specificity of the assay, with amplification successful on 79% of AHG scats tested in-field. While the assay was highly effective, scat DNA degradation and inhibitors limited detection in 21% of AHG samples, highlighting preservation challenges. This new assay has already been applied operationally in the field, providing early detection of AHG, and preventing potential introduction of this species into new areas.

**Data availability statement:** All data underlying the findings of this study are included within the manuscript and its Supporting information files. The ND2 DNA sequences generated in this study are publicly available in GenBank under accession numbers PQ390829–PQ390908.

**Funding:** This study was supported through Murdoch University grant MU-112020, and by the State Government of Victoria (Agriculture Victoria Research). This study was funded by the Gorgon Project. The Gorgon Project is operated by an Australian subsidiary of Chevron and is a joint venture of the Australian subsidiaries of Chevron (47.3%), ExxonMobil (25%), Shell (25%), Osaka Gas (1.25%), MidOcean Energy (1%), and JERA (0.417%). The funders had no role in study design, data collection and analysis, decision to publish, or preparation of the manuscript.

**Competing interests:** The authors have declared that no competing interests exist.

## Section 1: Introduction

Non-invasive genetic sampling (NGS) has been widely applied over the last twenty years by wildlife biologists [1]. NGS is when DNA is extracted from animal traces, such as from hair [2], shed skin [3], faecal scats [2,4], or other sources, without handling, capturing or even observing the individual animal. NGS alleviates many of the difficulties associated with conventional identifications including morphological complexities, cryptic life stages, and globally declining taxonomic expertise. Applications of NGS include detection of rare or endangered species [5], conservation management [6], differentiating between morphologically similar species [1], population estimation (e.g., [7]), and population sex-ratios [8]. NGS has also been applied for monitoring of invasive species [4], however, studies using NGS as an early detection tool for biosecurity purposes remain rare.

One of the primary issues facing NGS of animals is the quality of trace DNA. Animal faecal scats are one of the most common NGS sources, however scats contain very little host animal DNA and considerable concentrations of inhibitors [1]. The diet of the animal can play a significant role in the success of DNA extraction from scats. For example, insect material in scats is known to have an inhibitory effect on amplification success when DNA concentrations of the target species are low [9]. Furthermore, scat samples are generally collected from wild animals whose scats are likely to have been exposed to a variety of weather conditions which can degrade the DNA prior to the scat being collected. Other factors, such as DNA extraction method and sample preservation, can also influence the success of extracting species DNA from scats [1]. These challenges highlight the need for more sensitive molecular techniques capable of detecting low-quality DNA under field conditions.

Loop-mediated isothermal amplification (LAMP) offers a promising alternative to conventional PCR based approaches. LAMP enables rapid amplification of target DNA with high specificity, sensitivity, and robustness under isothermal conditions [10]. Assays can be conducted under field conditions using portable equipment and provide robust results in less than one hour. Importantly, LAMP is relatively tolerant to common inhibitors, is highly efficient, and can sensitively detect low concentrations of target DNA (e.g., [11–14]), making it particularly suitable for use on NGS sources of DNA.

The Common House Gecko (*Hemidactylus frenatus* Dumeril & Bibron), also known as the Asian House Gecko (AHG), is one of the world's most successful invasive geckos [15,16]. In parts of its introduced range, *H. frenatus* has invaded natural habitats where impacts on native geckos have been demonstrated [17], or inferred [18,19]. This species poses an ongoing biosecurity risk; in New Zealand it accounts for up to 44% of all intercepted herpetofauna [20], and on a remote Australian island, it is the most frequently detected gecko intercepted at the border [21]. It presents particular risks to island ecosystems, which are often the last refuges for endangered species [22]. Detecting incursions of this species rapidly is important as populations can quickly achieve high densities that may preclude eradication [18].

Currently, *H. frenatus* detection relies primarily on auditory and visual surveys. Males produce distinctive calls [16], but these are limited to specific behavioural

contexts and environmental conditions, with typically only adult males calling in the presence of females, and only during warmer months around sunset and sunrise [16]. Visual surveys also require expertise to distinguish *H. frenatus* from morphologically similar native geckos (e.g., *Gehyra variagata* in Australia). Similarly, scats from different gecko species can appear almost identical. For these reasons, scat collections, combined with conventional PCR followed by DNA sequencing, are often used to confirm species presence [21]. However, such molecular methods are constrained by relatively long turnaround times (typically 2–4 weeks from sample collection to analysis), which limits their use as an early detection tool.

Development of a targeted LAMP assay for *H. frenatus* could provide a rapid, specific and field-deployable molecular tool capable of detecting AHG DNA from scats within one hour. Although LAMP is generally less sensitive than real-time PCR, its speed, simplicity, and inhibitor tolerance make it an attractive option for biosecurity surveillance. LAMP assays have already been developed successfully to aid early detection of several invasive invertebrate species, including fruit flies *Bactrocera tryoni* [23], *B. trivialis* [24], fall armyworm *Spodoptera frugiperda* [14], grape phylloxera *Daktulosphaira vitifoliae* [11], and honeybee ectoparasitic *Varroa* mites [13]. NGS LAMP assays have also been successfully developed and employed to detect traces of two invasive species, including one vertebrate species; shed larval skins of the Khapra beetle *Trogoderma granarium* [12], and traces of DNA of the red fox *Vulpes vulpes* that scavenged on harbour porpoise carcasses [25].

In this study, we developed and validated a novel LAMP assay for the detection of *H. frenatus* (AHG). We used NGS (from small reptile scats) and tissue samples to evaluate DNA assay specificity and sensitivity. Field testing was conducted on scats collected from (i) mainland Australia where *H. frenatus* is established, and (ii) Barrow Island, where the species is currently absent, but represents an ongoing incursion threat.

## Section 2: Materials and methods

### 2.1. Specimens examined

For developing, optimising, and testing the specificity of our LAMP assay, we used 26 tissue samples from the target species (AHG) together with 54 tissue samples from twenty-five Australian native gecko and skink species which co-occur geographically with AHG in Western Australia (Table 1). Samples were sourced by Chevron Australia, Helix Molecular Solutions (Helix), and the Western Australian Museum (WAM). One hundred and twenty lizard scat samples were used for optimising (n = 20) and field validating (n = 100) the assay, collected from multiple sites in Western Australia (see section 2.6 and S1 Table). Scat samples from Barrow Island were sourced by Murdoch University under animal ethics permit number R2859/16. Tissue samples were obtained from preserved specimens from the WAM, or through a Cadaver and Tissue Usage Notification form.

### 2.2. Optimisation of DNA extraction methods

**2.2.1. Laboratory DNA extractions.** A destructive DNA extraction method (i.e., involving complete sample homogenisation) was used on AHG samples to obtain "high-quality" DNA with a high yield, using multiple Qiagen (Germany) DNA extraction kits. DNA was extracted from AHG tissue samples (R109230 and R109520, Table 1) using the DNeasy Blood and Tissue extraction kit (Qiagen) to obtain high yield DNA for downstream application to conduct the AHG LAMP sensitivity test (see Results).

Laboratory DNA extractions from scat samples were optimised using a QIAmp® Fast DNA Stool Mini kit following the manufacturers' protocols. Depending on the size of each scat a half or one-third of the scat was used in the DNA extraction with the remaining fraction stored for later use at −20 °C. A panel of DNA was destructively extracted from twenty AHG scat samples (collected from Dampier, Western Australia), using the extraction kit, which was found to produce DNA with a reasonably high yield. Two of these AHG scat samples were processed for downstream application to conduct the AHG LAMP sensitivity testing.

**Table 1. Gecko and skink DNA samples (from tissue) used for development and initial testing of AHG LAMP assay.**

| | Species | Sample | Source | Extraction method | LAMP Time (min) | LAMP Anneal (°C) | GenBank Accession (ND2) |
|---|---|---|---|---|---|---|---|
| 1 | *Hemidactlyus frenatus* | BS01 Karratha, Australia | Helix | Qiagen | 9.25 | 85.5 | PQ390829 |
| 2 | *Hemidactlyus frenatus* | BS10 Karratha, Australia | Helix | Qiagen | 9.00 | 85.5 | PQ390830 |
| 3 | *Hemidactlyus frenatus* | BS26 Batam, Indonesia | Helix | Qiagen | 12.25 | 85.5 | PQ390831 |
| 4 | *Hemidactlyus frenatus* | BS28 Batam, Indonesia | Helix | Qiagen | 15.00 | 86.2 | PQ390832 |
| 5 | *Hemidactlyus frenatus* | HR18 Intercepted | Helix | Qiagen | 8.50 | 85.3 | PQ390833 |
| 6 | *Hemidactlyus frenatus* | HR19 Intercepted | Helix | Qiagen | 8.50 | 85.2 | PQ390834 |
| 7 | *Hemidactlyus frenatus* | R131793, Kununurra, Australia | WAM | QE | 7.25 | 85.3 | PQ390835 |
| 8 | *Hemidactlyus frenatus* | R131879, Kununurra, Australia | WAM | QE | 7.50 | 85.2 | PQ390836 |
| 9 | *Hemidactlyus frenatus* | R132949, Broome, Australia | WAM | QE | 9.75 | 85.3 | PQ390837 |
| 10 | *Hemidactlyus frenatus* | R132952, Broome, Australia | WAM | QE | 10.00 | 85.3 | PQ390838 |
| 11 | *Hemidactlyus frenatus* | R164914, Truscott Airstrip, Australia | WAM | QE | 13.25 | 85.4 | PQ390839 |
| 12 | *Hemidactlyus frenatus* | R172512, Intercepted | WAM | QE | 16.25 | 85.8 | PQ390840 |
| 13 | *Hemidactlyus frenatus* | R172517, Intercepted | WAM | QE | 9.25 | 85.3 | PQ390841 |
| 14 | *Hemidactlyus frenatus* | R172557, Intercepted | WAM | QE | 11.25 | 85.3 | PQ390842 |
| 15 | *Hemidactlyus frenatus* | R174322, Port Hedland, Australia | WAM | QE | 7.50 | 85.1 | PQ390843 |
| 16 | *Hemidactlyus frenatus* | R174747, Carnarvon, Australia | WAM | QE | 7.50 | 85.3 | PQ390844 |
| 17 | *Hemidactlyus frenatus* | R173044, Cocos (Keeling) Island | WAM | QE | 10.75 | 85.1 | PQ390845 |
| 18 | *Hemidactlyus frenatus* | R173054, Cocos (Keeling) Island | WAM | QE | 13.25 | 85.1 | PQ390846 |
| 19 | *Hemidactlyus frenatus* | R112253, Yamdena Island, Indonesia | WAM | QE | 11.75 | 85.6 | PQ390847 |
| 20 | *Hemidactlyus frenatus* | R109929, Yamdena Island, Indonesia | WAM | QE | 13.25 | 85.6 | PQ390848 |
| 21 | *Hemidactlyus frenatus* | R109377, Banda island, Indonesia | WAM | QE | 11.50 | 85.2 | PQ390849 |
| 22 | *Hemidactlyus frenatus* | R109370, Banda island, Indonesia | WAM | QE | 6.50 | 85.4 | PQ390850 |
| 23 | *Hemidactlyus frenatus* | R109230, Nusa Penida Island, Indonesia | WAM | Qiagen | 8.00 | 85.4 | PQ390851 |
| 24 | *Hemidactlyus frenatus* | R109517, Wokam island, Indonesia | WAM | QE | 7.00 | 85.4 | PQ390852 |
| 25 | *Hemidactlyus frenatus* | R109520, Wokam island, Indonesia | WAM | Qiagen | 8.25 | 85.5 | PQ390853 |
| 26 | *Hemidactlyus frenatus* | R109879, Ambon Island, Indonesia | WAM | QE | 6.50 | 85.4 | PQ390854 |
| 27 | *Gehyra pilbara* | EA01 R165746, Australia | Helix | Qiagen | N/A | N/A | PQ390855 |
| 28 | *Gehyra pilbara* | EA02 R165753, Australia | Helix | Qiagen | N/A | N/A | PQ390856 |
| 29 | *Gehyra variegata* | EA03 R154142, Australia | Helix | Qiagen | N/A | N/A | PQ390857 |
| 30 | *Gehyra variegata* | EA04 R165763, Australia | Helix | Qiagen | N/A | N/A | PQ390858 |
| 31 | *Heteronotia binoei* | EA07 R154100, Australia | Helix | Qiagen | N/A | N/A | PQ390859 |
| 32 | *Heteronotia binoei* | EA08 R154114, Australia | Helix | Qiagen | N/A | N/A | PQ390860 |
| 33 | *Lucasium stenodactylum* | EA09 T20121209.HPZ2-03lucste, Australia | Helix | Qiagen | N/A | N/A | PQ390861 |
| 34 | *Strophurus jeanae* | EA11 R154145, Australia | Helix | Qiagen | N/A | N/A | PQ390862 |
| 35 | *Carlia triacantha* | EA12 T20130323.22HFZ2-03CARTRI.01, Australia | Helix | Qiagen | N/A | N/A | PQ390863 |
| 36 | *Carlia triacantha* | EA13 T20130412.23HPZ2-02CARTRI.01, Australia | Helix | Qiagen | N/A | N/A | PQ390864 |
| 37 | *Cryptoblepharus plagiocephalus* | EA14 R154110, Australia | Helix | Qiagen | N/A | N/A | PQ390865 |
| 38 | *Cryptoblepharus plagiocephalus* | EA15 R154113, Australia | Helix | Qiagen | N/A | N/A | PQ390866 |
| 39 | *Ctenotus grandis* | EA16 R154081, Australia | Helix | Qiagen | N/A | N/A | PQ390867 |
| 40 | *Ctenotus grandis* | EA17 R154084, Australia | Helix | Qiagen | N/A | N/A | PQ390868 |

*(Continued)*

**Table 1.** (Continued)

| | Species | Sample | Source | Extraction method | LAMP Time (min) | LAMP Anneal (°C) | GenBank Accession (ND2) |
|---|---|---|---|---|---|---|---|
| 41 | *Ctenotus Pantherinus acripes* | EA18 R154151, Australia | Helix | Qiagen | N/A | N/A | PQ390869 |
| 42 | *Ctenotus Pantherinus acripes* | EA19 R154168, Australia | Helix | Qiagen | N/A | N/A | PQ390870 |
| 43 | *Ctenotus saxatilis* | EA20 R154092, Australia | Helix | Qiagen | N/A | N/A | PQ390871 |
| 44 | *Ctenotus saxatilis* | EA21 R154099, Australia | Helix | Qiagen | N/A | N/A | PQ390872 |
| 45 | *Ctenotus serventyi* | EA22 T20120521.HPZ2A-06.01, Australia | Helix | Qiagen | N/A | N/A | PQ390873 |
| 46 | *Cyclodomophus melanops* | EA23 R154130, Australia | Helix | Qiagen | N/A | N/A | PQ390874 |
| 47 | *Cyclodomophus melanops* | EA24 R154160, Australia | Helix | Qiagen | N/A | N/A | PQ390875 |
| 48 | *Eremiascincus richardsonii* | EA25 R165786, Australia | Helix | Qiagen | N/A | N/A | PQ390876 |
| 49 | *Eremiascincus richardsonii* | EA26 T20121113.OFZ1-02ERERIC.01, Australia | Helix | Qiagen | N/A | N/A | PQ390877 |
| 50 | *Eremiascincus isolepis* | EA27 T20120519.HFZ1-03.01, Australia | Helix | Qiagen | N/A | N/A | PQ390878 |
| 51 | *Eremiascincus isolepis* | EA28 T20120520.HFZ2-02.01, Australia | Helix | Qiagen | N/A | N/A | PQ390879 |
| 52 | *Lerista bipes* | EA29 R154118, Australia | Helix | Qiagen | N/A | N/A | PQ390880 |
| 53 | *Lerista bipes* | EA30 R154121, Australia | Helix | Qiagen | N/A | N/A | PQ390881 |
| 54 | *Lerista clara* | EA31 R154140, Australia | Helix | Qiagen | N/A | N/A | PQ390882 |
| 55 | *Lerista clara* | EA32 R154166, Australia | Helix | Qiagen | N/A | N/A | PQ390883 |
| 56 | *Metetia greyii* | EA33 R154144, Australia | Helix | Qiagen | N/A | N/A | PQ390884 |
| 57 | *Metetia greyii* | EA34 R154173, Australia | Helix | Qiagen | N/A | N/A | PQ390885 |
| 58 | *Morethia ruficauda* | EA35 R154174, Australia | Helix | Qiagen | N/A | N/A | PQ390886 |
| 59 | *Notoscincus ornatus* | EA37 R154137, Australia | Helix | Qiagen | N/A | N/A | PQ390887 |
| 60 | *Notoscincus ornatus* | EA38 R154147, Australia | Helix | Qiagen | N/A | N/A | PQ390888 |
| 61 | *Proablepharus reginae* | EA39 R154135, Australia | Helix | Qiagen | N/A | N/A | PQ390889 |
| 62 | *Proablepharus reginae* | EA40 R154136, Australia | Helix | Qiagen | N/A | N/A | PQ390890 |
| 63 | *Ctenotus duricola* | EA41 T20131114.28HPZ2-01.CTEDUR.01, Australia | Helix | Qiagen | N/A | N/A | PQ390891 |
| 64 | *Ctenotus duricola* | EA42 T20131106.28HPZ2-04.CTEDUR.01, Australia | Helix | Qiagen | N/A | N/A | PQ390892 |
| 65 | *Lucasium stenodactylum* | EA43 T20131115.28GPZ1-08.LUCSTE.01, Australia | Helix | Qiagen | N/A | N/A | PQ390893 |
| 66 | *Morethia ruficauda* | EA44 T20131110.28HPZ2-01.MORRUF.01, Australia | Helix | Qiagen | N/A | N/A | PQ390894 |
| 67 | *Lerista elegans* | EA45 R165983, Australia | Helix | Qiagen | N/A | N/A | PQ390895 |
| 68 | *Lerista elegans* | EA46 R165978, Australia | Helix | Qiagen | N/A | N/A | PQ390896 |
| 69 | *Morethia lineoocellata* | EA47 R165977, Australia | Helix | Qiagen | N/A | N/A | PQ390897 |
| 70 | *Morethia lineoocellata* | EA48 R123950, Australia | Helix | Qiagen | N/A | N/A | PQ390898 |
| 71 | *Ctenotus hanloni* | EA49 R157543, Australia | Helix | Qiagen | N/A | N/A | PQ390899 |
| 72 | *Ctenotus hanloni* | EA50 R157557, Australia | Helix | Qiagen | N/A | N/A | PQ390900 |
| 73 | *Ctenotus serventyi* | EA51 R102812, Australia | Helix | Qiagen | N/A | N/A | PQ390901 |
| 74 | *Gehyra* sp. | EA55 R165742, Australia | Helix | Qiagen | N/A | N/A | PQ390902 |
| 75 | *Gehyra* sp. | EA58 R163105, Australia | Helix | Qiagen | N/A | N/A | PQ390903 |
| 76 | *Gehyra* sp. | EA59 R163044, Australia | Helix | Qiagen | N/A | N/A | PQ390904 |
| 77 | *Gehyra* sp. | EA60 R165748, Australia | Helix | Qiagen | N/A | N/A | PQ390905 |
| 78 | *Gehyra* sp. | EA61 R165749, Australia | Helix | Qiagen | N/A | N/A | PQ390906 |

*(Continued)*

**Table 1.** (Continued)

| | Species | Sample | Source | Extraction method | LAMP Time (min) | LAMP Anneal (°C) | GenBank Accession (ND2) |
|---|---|---|---|---|---|---|---|
| 79 | *Gehyra* sp. | EA63 R165255, Australia | Helix | Qiagen | N/A | N/A | PQ390907 |
| 80 | *Strophurus jeanae* | FD01 T20131203.29GPZ1-06.STRJEA.01, Australia | Helix | Qiagen | N/A | N/A | PQ390908 |

AHG samples are indicated by grey shading. "N/A" indicates no amplification produced with the new AHG LAMP assay, QE = QuickExtract, Qiagen = DNeasy extraction, Helix = Helix Molecular Solutions, WAM = Western Australian Museum, ND2 = mitochondrial ND2 locus.

**2.2.2. In-field compatible DNA extractions.** Optimising an in-field compatible DNA extraction procedure is a prerequisite for conducting the LAMP assay in the field. Non-destructive in-field DNA (i.e., not involving complete sample homogenisation) was extracted from three scat samples (collected from Dampier, WA) using commercial QuickExtract™ DNA extraction (QE) solution 1.0 (Epicentre, USA). This method is rapid and easy to use, providing good quality DNA for use as a template in the LAMP assay. One hundred microliters of QE solution were pipetted into 6-wells of each Genie strip (OptiGene, UK). Depending on the size of the scat only one-third or half of it was transferred into each individual well and broken using a toothpick (single use to prevent cross contamination) [11], with up to six samples processed simultaneously in the Genie III (OptiGene, UK). The in-field QE DNA extraction protocol used the Genie III as an incubator programmed at 65 °C for 6 min followed by 98 °C for 2 min (total = 8 min), following published protocols [11]. To experimentally test the effect of reducing possible inhibitors, two microliters of undiluted QE DNA extracted from scats (1:1) was diluted, 1:10, 1:20 and 1:50, in water for use as a template in the LAMP assay.

Additional, non-destructive in-field DNA samples were extracted from 18 AHG tissue (WAM) samples using fifty microliters of QE solution (Table 1). Tissue samples were removed from ethanol and air dried on paper towel for approximately one minute before being transferred into each well of the Genie strip using a toothpick (single use, to prevent cross contamination). Each sample was immersed in QE, with up to six samples processed simultaneously in the Genie III, as above. All DNA samples were stored at −20 °C for future use.

## 2.3. Assessment of mitochondrial ND2 DNA sequence variation

Lizard species identifications were confirmed through DNA barcoding of the mitochondrial NADH dehydrogenase subunit 2 (ND2) locus, following published PCR conditions [26], with the PCR annealing temperature reduced to 56 °C. Most AHG ND2 sequences generated in the current study were obtained using the new AHG LAMP primer "AHG_F3" (Table 2) in combination with the existing primer M1123R, 5'- GCTTAATTAAAGTGTYTGAGTTGC-3' [26] (~450 bp amplicon), for *Hemidactylus frenatus*. ND2 was amplified from other lizard species using a new 'general' primer "GeckoND2-F" designed to be applicable to this suite of small lizard species 5'- MCAAACMCGAAAAATYATRGC-3', in combination with M1123R (~570 bp amplicon). PCR's consisted of 40 cycles with an annealing temperature of 53 °C. PCR amplicons were sequenced commercially by Macrogen Inc. (Macrogen, Seoul, Korea) or Australian Genome Research Facility (AGRF) in dual directions.

DNA sequence electropherograms were analysed using the software MEGA7 [27]. Forward and reverse sequences were paired, and a consensus sequence was generated. Genetic similarity of DNA sequences of AHG and non-target lizard species was examined using a maximum likelihood tree constructed in Mega7 [27]. Pairwise sequence differences (p-distances) between species were generated in Mega7 [27] to assess percentage sequence divergences between species. Relationships among AHG samples were also examined through a haplotype network. DNA sequence variation in AHG specimens was examined through generation of a median joining haplotype network (ε = 0) [28], constructed in PopArt [29], using all new AHG sequences obtained here (Table 1, S1 Table) and from sequences of specimens from around the world obtained from GenBank [30].

**Table 2. AHG LAMP primer and amplicon sequences (gBlock) and parameters.**

| Primers | Sequence 5'-3' | Primer length (bp) | Predicted Tm, annealing temperature (°C) | Degeneracy of primer (fold) |
|---|---|---|---|---|
| AHG_F3 | CRCCACAGCCATATTCAYA | 19 | 55.9 | 4 |
| AHG_B3 | GGTAAGCTTGCGGCRAG | 17 | 59.0 | 2 |
| AHG_FIP | CCAAGTGACATAAGGGTAAYTATTA**CACCACAACCAARACCATTC** | 45 | 76.4 | 4 |
| AHG_BIP | CTAACAGGRTTYATGCCAAAAT**CGCYGATKGGYAGGAG** | 38 | 73.5 | 32 |
| AHG_Floop | GGGTTAGGGCTGGTGATAKTA | 21 | 66.9 | 2 |
| AHG_Bloop | KACTTATCATTAAAGACCTYGC | 22 | 46.1 | 4 |
| gBlock Fragment | cccCACCACAGCCATATTCACACACCACAACCAAGACCATTCcccTACTATCACCAGCCCTA-ACCCcccTAATAATTACCCTTATGTCACTTGGcccCTAACAGGGTTTATGCCAAAATcccGACT-TATCATTAAAGACCTTGCcccCTCCTACCAATCGGCGcccCTCGCCGCAAGCTTACCccc | 186 | N/A | N/A |

The F2 and B2 primer regions of FIP and BIP are in bold. Lowercase letters in the gBlock indicate extra "ccc" added between LAMP primer sites to increase the overall Tm of the amplicon.

## 2.4. Development and evaluation of AHG LAMP assay

**2.4.1. AHG dataset and primer design.** AHG LAMP primers were developed for a section of the mitochondrial ND2 locus that has been commonly used for DNA barcoding identification of geckos [26,30]. This locus has been well characterised from multiple *Hemidactylus* species, including AHG which is known to comprise multiple genetically distinct ND2 clades [30]. Fifty-four ND2 DNA sequences for non-target gecko and skink species that co-occur in Western Australia (samples 27–80, Table 1) were generated (as in section 2.3). These were combined with thirty-one DNA sequences of AHG (same region of ND2) downloaded from GenBank, mostly derived from [30]. The combined ND2 alignment was used to design eight novel LAMP primer regions for six primers to target AHG (Table 2), including two inner primers (FIP and BIP, each comprised of two primer regions, F1 + F2 and B1 + B2 respectively), two outer primers (F3 and B3), and two loop primers Floop and Bloop. For all primers the GC content (%), predicted melting temperature (Tm), and potential secondary structure (hairpins or dimers) were analysed using the Integrated DNA Technologies (IDT) online OligoAnalyzer Tool [31] using the qPCR parameter sets.

**2.4.2. LAMP primer ratio optimisation.** The primer ratio (F3/B3: FIP/BIP: Floop/Bloop) for this assay was optimised following published protocols using multiple ratios of primers [23]. Optimised primer master mix was prepared by adding the specified amount of each of the six primers in a 1:6:3 ratio. A 100 µL volume of primer master mix 1:6:3 (F3/B3: FIP/BIP: Floop/Bloop) was prepared by adding 10 µL (10 µM) each of F3/B3, 6 µL (100 µM) each of FIP/BIP, 3 µL (100 µM) each of Floop/Bloop and 62 µL of Ultrapure water (Invitrogen, Australia).

**2.4.3. LAMP assay conditions.** The AHG LAMP assay was performed following published protocols [14]. Each LAMP reaction was conducted in a total volume of 25 µL containing 14 µL of Isothermal Master Mix (Iso-001, OptiGene, UK), 10 µL of the primer master mix (as in section 2.4.2) and 1 µL of template DNA. A one microliter disposable plastic loop was used to add in-field DNA extract, ensuring that a fresh loop was used for each sample. The use of disposable plastic loops has been shown to be an effective method for transferring template DNA in LAMP reactions [23]. These loops reduce the risk of cross contamination and eliminate the need for pipettes when LAMP is used outside a laboratory, a necessity for effective field deployment.

High-quality DNA (Qiagen) extracted from AHG tissue samples was used as a positive control for optimising the LAMP assay. A no template control (NTC) was included in each test to detect reagent contamination. All the LAMP assays were run in the Genie III at 65 °C for 25 min followed by an annealing curve analysis from 98 °C to 73 °C with ramping at 0.05 °C/s. The total run time was approximately 35 minutes. The amplification and anneal derivative temperature profiles

(anneal derivative) were visualised on the Genie III screen to ensure that amplification occurred as expected. Positive results were further confirmed through the annealing step resulting in a single product peak at a specific temperature, while the negative (NTC) and non-target lizard DNA remained as a flat line. Results were analysed using a PC version of the software Genie Explorer version 2.0.7.11 in the blue channel.

**2.4.4. Evaluation of a gBlock gene fragment for AHG LAMP assay.** Gene Fragment (gBlocks) are targeted synthetic oligonucleotides originally developed to be used as standards in real-time PCR reactions [32] but are equally applicable to LAMP assays as controls to monitor assay performance as synthetic positive controls [14]. To develop a consistent positive control for use in the LAMP assay, a 186 bp gBlock gene fragment (Integrated DNA Technologies, Iowa, USA), modified from the complete fragment used for the assay, was designed for use as a synthetic DNA positive control. This synthetic fragment consisted solely of concatenated LAMP primers separated by runs of "ccc", to increase the overall Tm of the fragment (Table 2). To evaluate detection sensitivity, the copy number was determined, and a ten-fold serial dilution (1:10) of the gBlock was prepared as outlined in [11]. Sensitivity of the serially diluted gBlock was tested in the Genie III following the same AHG LAMP assay amplification conditions (as in section 2.4.3) with the amplification time increased from 25 minutes to 35 minutes, with this longer time included to allow detection of the gBlock present at very low concentrations. Following this, another LAMP run was conducted to determine the best dilution to be used as a positive control in the LAMP assay. The same five-fold serial dilution (10 ng/μL to 3.2 x 10$^{-3}$ ng/μL) of high-quality AHG tissue DNA (R109520, used previously) with up to six dilutions was used as a template to compare amplification time with a standard amount of one million copies (1 x 10$^5$ copies/μL) of gBlock.

## 2.5. Analytic sensitivity of the AHG LAMP assay

To test the sensitivity of the LAMP assay a five-fold serial dilution of high-quality DNA extract of two biological replicates of AHG scat samples (collected from Dampier, Western Australia) and two biological replicates of AHG tissue samples (R109230 and R109520, Table 1) were prepared using Ultrapure water (Invitrogen, Life Technologies, Australia). Starting DNA concentrations of all four samples were quantified using a Qubit 2.0 Fluorometer (Invitrogen, Life Technologies, Australia) following the manufacturers protocol. All four AHG DNA samples were serially diluted (total eight dilutions) from 10 ng/μL to 1.28 x 10$^{-4}$ ng/μL (1:1–1:78125). Sensitivity of the LAMP assay was tested using the serially diluted DNA in the Genie III following the same AHG LAMP assay amplification conditions (as in section 2.4.3). The time of amplification and anneal derivative temperature were recorded for all 8 dilutions. To further explore any differences between DNA obtained from AHG tissue compared with DNA obtained from scats LAMP amplification times and anneal derivative temperature were compared using Student's T-tests in Excel (365 MSO, Version 2503).

## 2.6. Specimens examined for field testing and lab validation of field samples

In total, 108 samples were tested for the AHG LAMP assay in the field. These samples consisted of 100 scats (S1 Table) collected from multiple Western Australian field locations: Barrow Island (n = 49), Dampier (n = 17), Karratha (n = 33) and Perth (n = 1); as well as eight tissue samples collected from Karratha, Australia (n = 4) and Batam, Indonesia (n = 4). DNA was extracted from both scats and tissue samples using the in-field QE DNA extraction method using the Genie III. The AHG LAMP assay was performed (as in section 2.4.3). Amplification time and anneal derivative temperature of all scats, tissues, gBlock (1 x 10$^6$ copies/μL) synthetic positive control, and negative no template control, were recorded.

All 108 non-destructive QE DNA extracts, and 100 vials containing the other two-thirds/ half of the field collected lizard scats were shipped to the laboratory on dry ice. These were then processed for AHG LAMP assay in the laboratory to validate AHG positive samples. Note that 1 μL loops were used to transfer template into master-mix for field testing whereas 1 μL pipette tips were used for adding template in the laboratory. High-quality DNA was extracted from the 100 dry lizard scat samples using the QIAamp® Fast DNA Stool Mini kit (Qiagen, Australia). These DNA extracts were used for further

confirmation for species identification through ND2 DNA barcoding (S1 Table) using methods outlined above (section 2.3), and primers GeckoND2-F/ M1123R.

## Section 3: Results

### 3.1. AHG LAMP assay design and optimisation

Novel LAMP primers were developed to target a 231 bp amplicon for AHG for the mitochondrial ND2 locus (Table 2, Fig 1). Assessment of the primer ratio (F3/B3: FIP/BIP: Floop/Bloop) for this assay found the optimal ratio to be 1:6:3, with final concentrations of 0.4 μM, 2.4 μM and 1.2 μM for the F3/B3, FIP/BIP and Floop/Bloop primers respectively.

### 3.2. DNA barcoding of specimens tested in the LAMP assay

Species-level identifications of lizard samples examined in the current study were confirmed through DNA barcoding of the ND2 locus (Table 3, Fig 2, S1 Table). Relationships among AHG samples were also examined using a haplotype network (Fig 3), which indicated that most Australian AHG samples were very closely related, with a small number of Australian and intercepted specimens being more genetically divergent. Interestingly, some of the Cocos Island (Australian territory) samples matched the most common Australian haplotype found, while others belonged to a highly divergent lineage (>6% different), that had not previously been genetically characterised ("Group E" on Fig 3).

### 3.3. DNA extractions

High-quality DNA extracted using the QIAamp® Fast DNA Stool Mini kit yielded enough DNA from AHG scats for downstream application, while the non-destructive DNA extraction method from scats and tissues using QuickExtract™ was

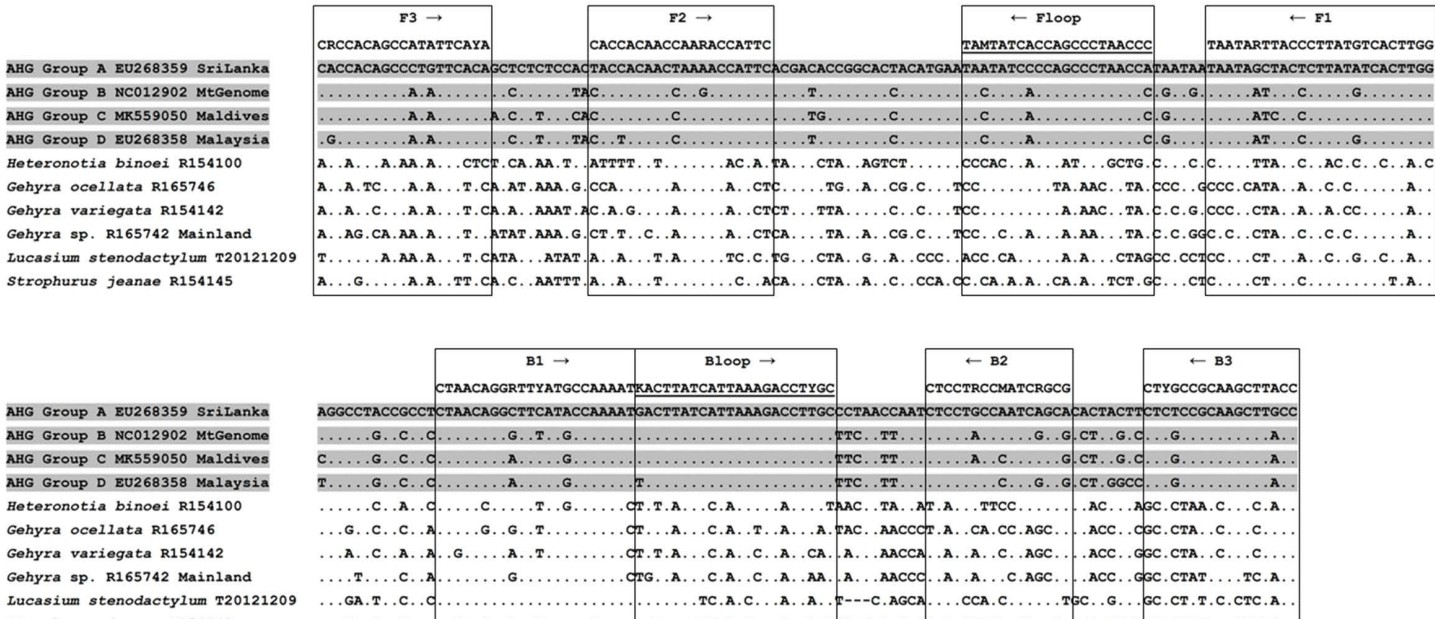

OFFICIAL

**Fig 1. AHG LAMP primer alignment.** Arrows indicate primer direction; Loop primers are underlined; FIP (5'-3') is made by combining F1 (reverse compliment) and F2; BIP (5'-3') is made by combining B1 (reverse compliment) and B2. Grey-shading indicates AHG, the target species, while non-target gecko species are unshaded. AHG DNA sequences were obtained from [30], while non-target DNA sequences were newly generated. AHG group letters (A-D) are from [30].

**Table 3. Field validation of AHG LAMP assay on lizard scat samples.**

| Species | Collected Mainland, Australia | Collected Barrow Island, Australia | LAMP Positive (in field) | LAMP Negative (in field) |
|---|---|---|---|---|
| *Hemidactlyus frenatus* | 38 | 0 | 30 | 8 |
| *Cryptoblepharus plagiocephalus* | 0 | 3 | 0 | 3 |
| *Gehyra pilbara* | 1 | 11 | 0 | 12 |
| *Gehyra* sp. | 1 | 0 | 0 | 1 |
| *Gehyra variegata* | 0 | 5 | 0 | 5 |
| *Heteronotia binoei* | 1 | 0 | 0 | 1 |

AHG samples indicated by grey shading. Species identifications were made through positive LAMP amplification and/or subsequent laboratory DNA barcoding of ND2 (see S1 Table).

found to be suitable for conducting LAMP assays in the field. All in-field QE DNA extracts from tissue samples, n = 18 (Table 1) amplified within the expected time range 10 ± 2.8 minutes (average ± SD). The anneal derivative temperature for all the samples was 85.3 ± 0.2 °C.

Additionally, three in-field QE DNA extracts from scat samples were tested both undiluted (1:1) and diluted at 1:10, 1:20 and 1:50 (Fig 4 and S2 Data). All four DNA dilutions amplified within the expected time range; with 1:1 in 11.75 ± 1.75 minutes, 1:10 in 12.3 ± 1.9 minutes, 1:20 in 16 ± 5.2 minutes and 1:50 in 11.9 ± 0.5 minutes. From these results we recommend using DNA at 1:1, because amplification times were not substantially shorter following dilution. All the samples tested produced an anneal derivative temperature at ~85.5 °C, which was not affected by DNA dilution.

### 3.4. Performance of the LAMP assay

The AHG LAMP assay produced amplification from AHG DNA from tissue samples in an average of 10.0 ± 2.7 minutes (Fig 5a and 5c, Table 1), with an anneal derivative temperature at 85.4 ± 0.2 °C (Fig 5b and 5d, Table 1); with amplification in under 25 minutes considered as positive. The non-template control in all the LAMP runs tested did not amplify confirming the absence of primer dimer, and that there were no false positive amplifications due to primer interactions or reagent contamination (Fig 5a and 5c).

"Blind" sample LAMP testing of the 108 tissue and scat samples was conducted in-field producing positive amplification only from AHG tissue and scats (n = 38, S1 Table). Subsequent laboratory testing showed eight additional AHG scats (21% of total scats identified as AHG) were not detected through the initial in-field DNA extractions by LAMP, with these samples requiring laboratory high-quality DNA extractions to produce enough DNA for secondary LAMP testing or DNA sequencing and barcoding identification (Table 3, S1 Table). Generally, 45% of the field collected lizard scats could not be amplified in the laboratory by PCR for DNA barcoding. Importantly, none of the non-target lizard scats (five other species) amplified in the field or laboratory, including all forty-nine scat samples tested from Barrow Island (S1 Table).

### 3.5. Evaluation of AHG gBlock gene fragment

The detection sensitivity of AHG gBlock gene fragment evaluated for templates ranging from ~100 million copies down to ~10 copies was quite high, with as low as 1 x 10 copies/µL detected within 15 minutes. From the amplification profile 100,000 copies (1 x $10^5$ copies/µL) of gBlock amplified in 10 minutes which equates to ~2 ng/µL of AHG DNA (Fig 5e). Based on the sensitivity test of AHG gBlock 1 x $10^6$ copies/µL of gBlock amplified in 8 minutes. The neat DNA concentration, prior to dilution (10 ng/µL), amplified in 8 minutes (Fig 5e). Hence the recommended gBlock dilution would be 1 x $10^6$ copies/µL for use as synthetic positive control for future LAMP runs. The anneal derivative profiles of LAMP amplicons show two very similar peaks, 85.4 °C for AHG DNA and 85.5 °C for gBlock (Fig 5f).

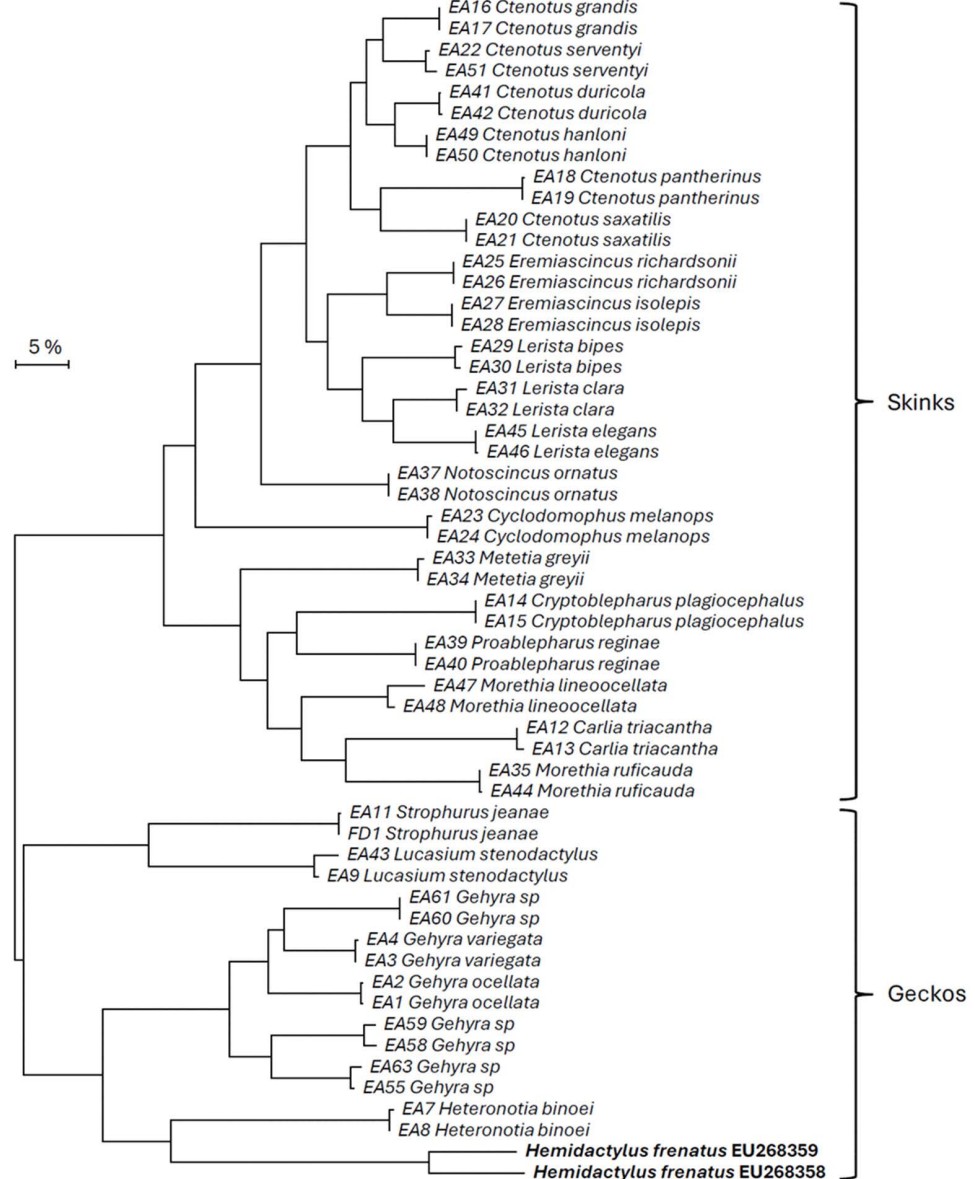

**Fig 2. Maximum-likelihood tree of relationships between *Hemidactylus frenatus* (AHG, in bold, GenBank reference sequences) with lizard species collected from Western Australia.**

The gBlock $1 \times 10^6$ copies/µL was used in all the LAMP runs as a synthetic positive control to monitor the performance of the assay in the field. Across eighteen LAMP runs the time for positive amplification was consistent, amplifying in $12 \pm 0.6$ minutes with an anneal derivative temperature of $85 \pm 0.3$ °C. Furthermore, the negative control which had no DNA template (NTC) did not amplify in any of the eighteen LAMP runs conducted, showing a flat line both in the amplification and anneal derivative profiles confirming the absence of contaminants in the reagent mix. The results of AHG LAMP assay from both the field testing and laboratory assays showed similar results, demonstrating the robustness of the assay.

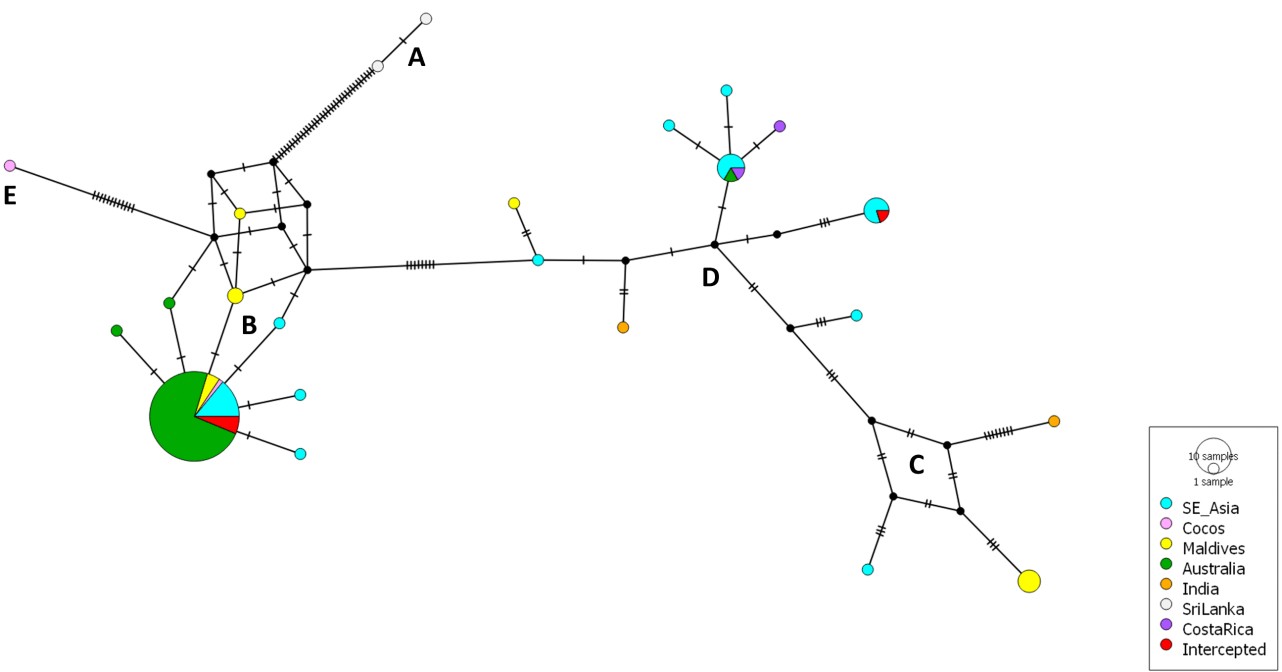

**Fig 3. Haplotype network of all AHG sequences obtained from GenBank and generated in the current study.** Letters refer to previously recognised genetic groups of AHG (following clade naming system in [30]), apart from 'E' (Cocos Island, territory of Australia), which is newly reported here. Australian samples examined in the current study are shown in green. Red samples were intercepted in transit to Barrow Island, Australia.

### 3.6. Sensitivity of LAMP assay

The sensitivity of the AHG LAMP assay was tested on five-fold serial dilutions of two biological replicates of each AHG tissue and scat DNA. The time to amplify DNA increased as DNA template concentrations reduced (Fig 6). All eight DNA dilutions from tissue samples produced positive amplification curves with the highest DNA concentration (10 ng/μL) amplifying in 8 minutes compared with the scat DNA which amplified in $11 \pm 1.4$ minutes (Fig 6). The LAMP test was sensitive in detecting DNA from tissue samples down to the lowest level $1.28 \times 10^{-4}$ ng/μL (0.28 picogram) of DNA in 22 minutes. Sensitivity was lower for DNA dilutions of scat samples, with amplification up to only four dilutions ($8 \times 10^{-2}$ ng/μL) in 14 minutes (Fig 6). T-tests (S2 Data) showed a significant ($p < 0.02$) difference between amplification times for DNA derived from tissue and from scats, while there was no significant difference observed for their anneal derivative profiles ($p > 0.25$).

### Section 4: Discussion

This study designed and optimised a new LAMP assay for the detection of an invasive vertebrate species. The assay was successfully applied to both tissue and scat AHG samples (Tables 1 and 3). These samples represented much of the known AHG genetic diversity (compared with ND2 sequences from [30]), with some specimens from AHG "Group E" being more than 6% genetically divergent from other groups (Fig 3). AHG is known to possess highly distinct genetic lineages, and likely represents a species complex [30,33]. The presence of two highly divergent genetic lineages amongst our samples indicates that there have likely been at least two independent introduction events to the Cocos (Keeling) Islands.

The new LAMP assay performed well on all samples tested including those from "Group E", allowing all positive AHG samples to be rapidly identified, apart from a small number of field-collected scat samples which yielded low amounts of DNA. Unfortunately, we did not have access to specimens of any congeneric species to experimentally test the performance of the AHG LAMP assay on other *Hemidactylus* species. Another closely related species, *Hemidactylus platyurus*,

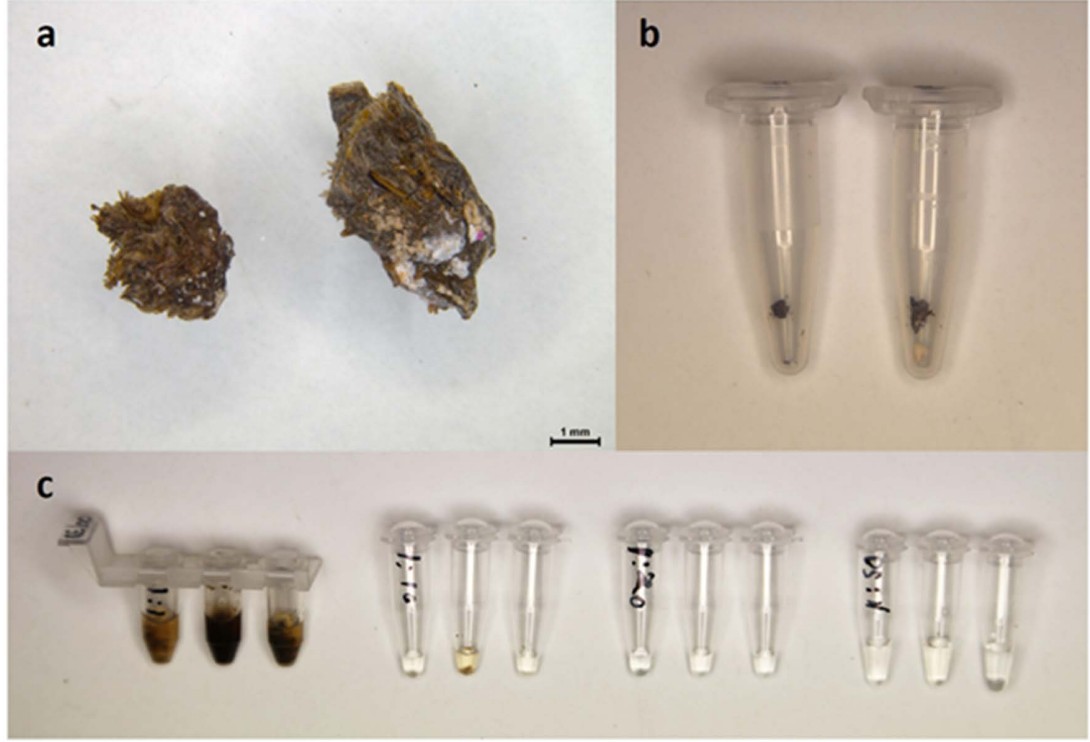

**Fig 4. Optimal method for obtaining in-field DNA extracts in the field. a)** Scats, 1/3 approx. 3 x 3 mm (left), 1/2 approx. 3 x 5 mm (right). **b)** The same scats in 1.5 ml tubes, pre-DNA extraction. **c)** Extracted DNA (100 µL QuickExtract™) (1:1) subsequently diluted (1:10, 1:20 and 1:50) using water.

occurs north of Australia, however this species (represented by GenBank EU268352) is 36% different for the section of ND2 used in the LAMP assay, so would be unlikely to amplify.

We designed and optimised a synthetic DNA positive control (gBlock) for use in the AHG LAMP assays following [11]. This synthetic DNA is beneficial in: (i) providing a consistent control to allow tracking of the performance of LAMP assays across runs, and (ii) providing a relatively high amount of control DNA compared with DNA extractions from gecko scat specimens. Scat samples in general produce very low DNA yields [1] due to the small amount of DNA contained within scats. DNA is present at low levels in scat samples compared to blood or tissue samples [34], and can be further reduced by unfavourable environmental conditions with DNA degrading over time [35]. Therefore, the freshest samples should be used when possible. If the scats are not assayed immediately, they can be preserved frozen (approx. −20 °C) to reduce the rate of DNA degradation. For field collected scat samples, the LAMP assay reduces the risk associated with storage of scats by providing an in-situ assay that can be completed immediately as required.

When working with scats, the risk of cross contamination is increased due to the low amount of target DNA contained within scats [34]. To reduce this risk, additional precautions should be taken when field collecting scats. The items used to collect samples should be disposable or washed and sterilised between each sample to avoid cross-contamination. Similarly, the amplicons produced during LAMP assays could result in potential cross contamination between samples if precautions, such as not opening LAMP reaction tubes post amplification, are not followed. For this reason, disposable items should also be used to avoid transferring amplified DNA into the pre-amplification working space, and negative no-template controls should always be included to test for possible LAMP reagent contamination [23].

Amplification of the LAMP assay was found to become slower in a predictable manner as DNA template concentrations reduced. At low DNA concentration of AHG tissue <0.001 ng/µl, the average LAMP amplification time was less than 15

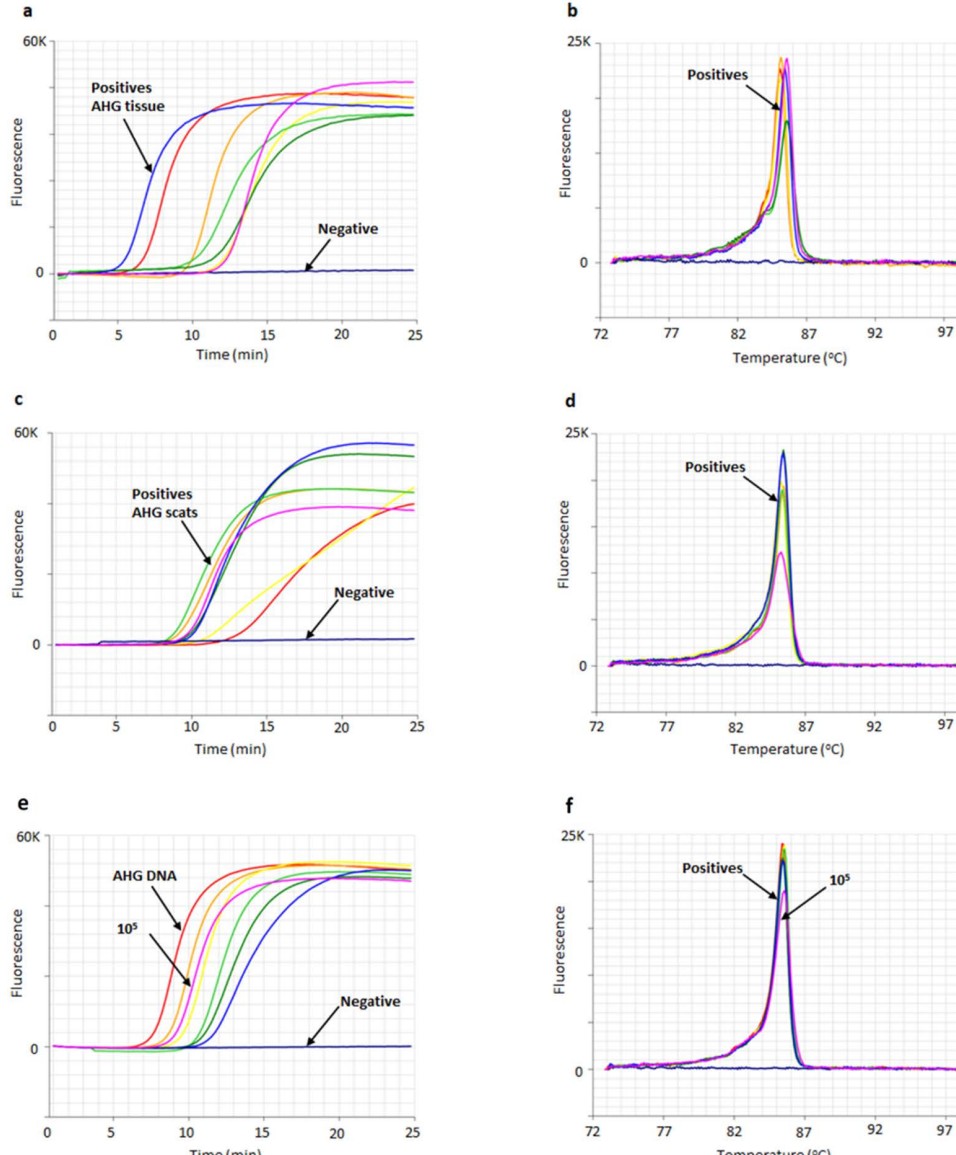

**Fig 5. Amplification and anneal derivative temperature profiles for the optimised AHG LAMP assay using 1:6:3 primer ratio. a)** Positive amplification profile of seven AHG tissue samples within 6 to 13 minutes (in-field QuickExtract DNA extracts). Negative, no amplification (flat line, blue). **b)** Anneal derivative temperature is ~85.4 °C for all 7 AHG tissue DNA. **c)** Positive amplification profile of 7 AHG scat samples within 10 to 14 minutes (high-quality Qiagen DNA extracts). Negative, no amplification (flat line, blue). **d)** Anneal derivative temperature for scat and tissue samples are the same at 85.4 °C. **e)** Amplification profile of five-fold dilution of AHG tissue DNA (R109520) up to 6 dilutions ranging from 10 ng/µL to $3.2 \times 10^{-3}$ ng/µL (amplification time from highest to lowest DNA concentration was 8 to 12 minutes) and gBlock $1 \times 10^5$ copies/µL, at 10 minutes (pink). Negative, no amplification (blue). **f)** Anneal derivative profiles of LAMP amplicons with similar peaks, 85.4 °C for AHG DNA and 85.5 °C for gBlock (pink).

minutes (Fig 6). Similarly, the average LAMP amplification time of gecko scat DNA at low concentrations, <1 ng/µl was also less than 15 minutes (Fig 6). The difference between the lower amplification threshold of tissue and scat samples is likely due to the presence of non-target DNA (e.g., bacteria, digested food etc.) in the scat samples overinflating the overall DNA quantification measurements, with scats also known to have degraded DNA and amplification inhibitors [1,35].

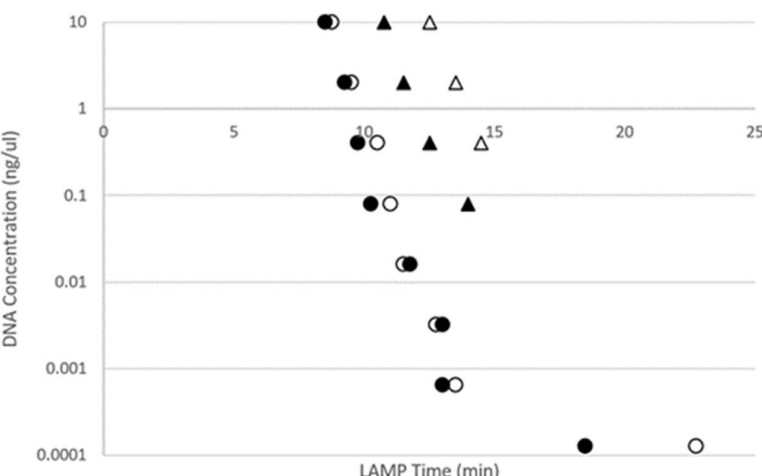

**Fig 6. AHG LAMP sensitivity test for a five-fold serial dilution of DNA samples from two AHG tissue (R109230 and R109520) (circles) and two AHG scat (from Dampier, Western Australia) (triangles).** Amplification times are shown for all 4 DNA samples with DNA concentration ranging from highest (10 ng/µL) to lowest (1.28 x 10$^{-4}$ ng/µL). All eight DNA dilutions of tissue samples amplified (10 ng/µL DNA amplified in 8 minutes and 1.28 x 10$^{-4}$ ng/µL in 22 minutes). DNA dilutions of scat samples only amplified up to four dilutions (10 ng/µl DNA amplified in 10 and 12 minutes and 8 x 10$^{-2}$ ng/µL in 14 minutes).

The amount of the scat used in DNA extraction also appears to influence the quality and quantity of DNA extracted. If the whole scat size were used (rather than the recommended one half or third), amplification times were found to increase, likely due to amplification inhibitors such as humic acid present in the scats, which is known to slow LAMP reactions [36]. Other molecular diagnostic approaches, such as DNA barcoding and real-time PCR, are known to be even more sensitive to the presence of inhibitors, and are unlikely to be successful in amplification from crude (scat) DNA extracts [1,36]. This, combined with the ease of use, and ability to rapidly amplify and detect positive targets in portable laboratory/ in-field compatible devices are the major advantages LAMP has over other molecular approaches [36].

The LAMP assay outlined here provides a reliable screening tool for the detection of AHG in the field, detecting 100% of AHG tissue samples, and 79% of field scat samples from traces (scats) of AHG. However, as when using any molecular approach, a positive AHG detection should always be verified, particularly if the detection is in an unusual or previously undocumented location. Verification can be performed through using standard DNA barcoding methods, as conducted here. Our study provides new ND2 DNA barcode sequences for an additional twenty-five gecko and skink species present in Western Australia, which greatly improves resources available for molecular identification of these small lizards in Australia. However, verification through DNA barcoding is relatively slow, requiring a turnaround time of days to weeks, compared with LAMP which can produces results in approximately one hour. Therefore, it may be preferable to verify positive LAMP results by increased sampling efforts, through LAMP confirmation of additional samples from the area the detection was made.

The AHG LAMP assay was operationalised in 2022 and is now widely used by Chevron Australia in their biosecurity operations in Australia. This new tool provides accurate rapid diagnosis of intercepted AHG (e.g., Table 1). In August 2023, a scat was collected that was suspected of originating from an AHG, detected on a half-height container during a quarantine inspection at the border of Barrow Island. Following in-situ analysis of the scat using the new AHG LAMP assay the scat was identified as originating from an AHG. A rapid decision was made to reject the cargo and vessel from Barrow Island. Rejecting cargo and vessels from any port is an expensive operation that is usually met with resistance from stakeholders. The AHG LAMP assay has reduced the subjectivity associated with this decision, which ultimately

enhanced cooperation from stakeholders, leading to a quick response. The scat was subsequently sent to Perth for validation of identification through DNA barcoding (as in section 2.3) with DNA sequencing results confirming the LAMP assay identification of an AHG with 100% confidence.

## Supporting information

**S1 Table. Samples tested in-field: n = 100 scats, n = 8 tissue samples.** Species identify determined through positive LAMP result and/or DNA barcoding.
(DOCX)

**S2 Data. T-test comparisons of LAMP amplification times and anneal derivatives from scat and tissue DNA samples.** LAMP optimisation runs for proportion of scat sample extracted, and scat DNA template dilutions in water (1:1, 1:10, 1:50).
(XLSX)

## Acknowledgments

Scat samples from Barrow Island were sourced by Murdoch University. Scats samples from Dampier were provided by Chevron Australia. Tissue samples were provided by both the Western Australian Museum and Helix Molecular Solutions.

## Author contributions

**Conceptualization:** Melissa L. Thomas.

**Data curation:** Arati Agarwal, Mark J. Blacket.

**Formal analysis:** Arati Agarwal, Mark J. Blacket.

**Funding acquisition:** Melissa L. Thomas, Simon J. McKirdy, Brendan C. Rodoni, Mark J. Blacket.

**Investigation:** Arati Agarwal, Yvette Hitchen, Mark J. Blacket.

**Methodology:** Arati Agarwal, Yvette Hitchen, Mark J. Blacket.

**Resources:** Yvette Hitchen, Paul Doughty.

**Supervision:** Arati Agarwal, Melissa L. Thomas, Yvette Hitchen, Mark J. Blacket.

**Writing – original draft:** Arati Agarwal, Melissa L. Thomas, Mark J. Blacket.

**Writing – review & editing:** Arati Agarwal, Melissa L. Thomas, Yvette Hitchen, Paul Doughty, Simon J. McKirdy, Brendan C. Rodoni, Mark J. Blacket.

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
