## [Decision Letter · Decision Letter 0]

12 May 2025

Dear Dr. Blacket,

Thank you for submitting your manuscript to PLOS ONE. After careful consideration, we feel that it has merit but does not fully meet PLOS ONE’s publication criteria as it currently stands. Therefore, we invite you to submit a revised version of the manuscript that addresses the points raised during the review process.

Please submit your revised manuscript by Jun 26 2025 11:59PM If you will need more time than this to complete your revisions, please reply to this message or contact the journal office at plosone@plos.org . A rebuttal letter that responds to each point raised by the academic editor and reviewer. You should upload this letter as a separate file labeled 'Response to Reviewers'.A marked-up copy of your manuscript that highlights changes made to the original version. You should upload this as a separate file labeled 'Revised Manuscript with Track Changes'.An unmarked version of your revised paper without tracked changes. You should upload this as a separate file labeled 'Manuscript'.

We look forward to receiving your revised manuscript.

Kind regards,

Mirna Alejandra Gonzalez-Gonzalez, PhD

Academic Editor

PLOS ONE

Journal Requirements:

2. In the online submission form, you indicated that your data is available only on request from a third party. Please note that your Data Availability Statement is currently missing the name contact details for the third party, such as an email address or a link to where data requests can be made. Please update your statement with the missing information.

Additional Editor Comments:

1. Please ensure that table numbers are updated to match their order of appearance in the text.

2. Table 2, please include the abbreviations for Helix and WAM in the table legend, as all necessary information for interpreting the table should be provided.

3. Table 2, please include numbering for each row to facilitate the identification of the 26 AHG tissue samples and the 54 tissue samples from other gecko and skink species.

4. Please include an approximate weight for the half and one-third portions of scat used in the DNA extraction.

5. Please review the title of the last column in Table 1, as the text is overlapping and unreadable (Degener...)

6. In Discussion section, include limitations of the study.

7. In Discussion section, compare your results to other published articles.

Reviewers' comments:

Reviewer's Responses to Questions

**Comments to the Author**

1. Is the manuscript technically sound, and do the data support the conclusions?

Reviewer #1: Yes

2. Has the statistical analysis been performed appropriately and rigorously?

Reviewer #1: Yes

3. Have the authors made all data underlying the findings in their manuscript fully available?

Reviewer #1: Yes

4. Is the manuscript presented in an intelligible fashion and written in standard English?

Reviewer #1: Yes

Reviewer #1: The manuscript is technically sound, with experiments conducted rigorously using established molecular techniques, appropriate controls (gBlock, NTCs, non-target species), sufficient replication (multiple samples/dilutions), and adequate sample sizes for its aims. The data strongly support the conclusions, with clear evidence of the assay’s performance and application. Minor weaknesses—variable scat sizes, lack of statistical tests, and untested congeners—do not undermine the study’s validity but suggest areas for refinement. The conclusions are appropriately drawn, with limitations acknowledged, ensuring scientific integrity.

The descriptive approach is partially appropriate given the study’s focus on tool development and validation. The assay’s performance is convincingly demonstrated through means, SDs, and success rates, aligning with the objective of establishing functionality. However, the lack of inferential statistics limits the ability to rigorously confirm observed differences (e.g., tissue vs. scat sensitivity), which could strengthen claims about assay robustness.

The statistical analysis is technically appropriate but not rigorously applied. The descriptive statistics (means ± SD, percentages) effectively support the assay’s sensitivity (0.0001 ng/μL), specificity (100% non-target exclusion), and field utility (79% scat detection), aligning with the study’s goals. However, the absence of inferential statistics limits the ability to statistically validate observed differences, reducing rigor. For PLOS ONE, where methodological soundness is key, the current approach is sufficient for a methods paper, as the data’s quality and presentation carry the conclusions. Adding basic tests (e.g., t-tests) would enhance rigor without altering the study’s core findings. The analysis is adequate but could be improved with minimal statistical tests to confirm key trends, particularly for sensitivity and field performance comparisons.

The authors have made all data underlying their findings fully available, as required by PLOS ONE. The combination of: (1) Detailed tables (1-3) and figures (1-6) in the manuscript; (2) A comprehensive S1 Table (assumed to include raw field data); (3) Publicly accessible ND2 sequences in GenBank (PQ390829-PQ390908), (4) meets the policy’s standards for transparency and replicability.

Minor gaps (e.g., raw dilution times, scat metadata) are likely addressed in S1 Table, though confirmation requires reviewing the file. If S1 Table lacks these specifics, a slight revision (e.g., adding a supplementary table with dilution and scat details) would ensure completeness. The data availability is robust and compliant, with all critical findings supported by accessible data. Any small omissions are non-critical and can be easily rectified if needed.

I recommend acceptance with minor revisions. The study is methodologically sound, with significant practical value for biosecurity. To enhance its impact and meet PLOS ONE’s standards, please address the following:

(a) Clarify scat DNA limitations (e.g., 21% failure rate) in the abstract and Discussion.

(b) Justify methodological choices (e.g., ND2, scat size, dilutions) with data or citations.

(c) Quantify assay advantages over PCR (e.g., time, cost) in the Introduction and Discussion.

(d) Streamline figures (e.g., Figure 2) and ensure consistency (e.g., 25 vs. 35 minutes).

(e) Highlight the Cocos Island “Group E” finding as a novel contribution.

These revisions will strengthen the manuscript’s clarity, rigor, and appeal to a broad readership.

**Do you want your identity to be public for this peer review?** For information about this choice, including consent withdrawal, please see our Privacy Policy

---

## [Author Response · Author response to Decision Letter 1]

11 Jun 2025

Response to Reviewers

Journal Requirements:

DONE.

2. In the online submission form, you indicated that your data is available only on request from a third party. Please note that your Data Availability Statement is currently missing the name contact details for the third party, such as an email address or a link to where data requests can be made. Please update your statement with the missing information.

DONE. All data has been supplied in the manuscript, the Supplementary files, & GenBank.

DONE. “S1 Supplementary Table” legend is included at end of manuscript. An additional “S2 Supplementary Data” file has now been added. This additional file includes the T-test data (tissue verses scat comparison) as well as the scat DNA optimisation results (portion extracted and dilution comparisons), as noted below.

DONE.

Additional Editor Comments:

1. Please ensure that table numbers are updated to match their order of appearance in the text.

DONE. Table 1 and Table 2 have been renumbered to match the order in the text.

2. Table 2, please include the abbreviations for Helix and WAM in the table legend, as all necessary information for interpreting the table should be provided.

DONE. Added to legend (Table 1).

3. Table 2, please include numbering for each row to facilitate the identification of the 26 AHG tissue samples and the 54 tissue samples from other gecko and skink species.

DONE. Note, these numbers are also now used in the text to refer to non-target lizards (i.e., samples 27 to 80, Table 1).

4. Please include an approximate weight for the half and one-third portions of scat used in the DNA extraction.

DONE. Scats were not weighed for this study. Scat weight would depend heavily on the water content of scats. Instead, an indication of scat size has been added to the legend in Figure 4 (1/3 scat = approx. 3 x 3 mm, ½ scat = approx. 3x 5 mm).

5. Please review the title of the last column in Table 1, as the text is overlapping and unreadable (Degener...)

DONE. All data is present in the Table. The text issue appears to have occurred during PDF generation.

6. In Discussion section, include limitations of the study.

DONE. Additional discussion of limitations has been added (as outlined in the specific comments below).

7. In Discussion section, compare your results to other published articles.

DONE. References have been added throughout the Discussion.

Reviewer #1 Comments:

The manuscript is technically sound, with experiments conducted rigorously using established molecular techniques, appropriate controls (gBlock, NTCs, non-target species), sufficient replication (multiple samples/dilutions), and adequate sample sizes for its aims. The data strongly support the conclusions, with clear evidence of the assay’s performance and application. Minor weaknesses—variable scat sizes, lack of statistical tests, and untested congeners—do not undermine the study’s validity but suggest areas for refinement. The conclusions are appropriately drawn, with limitations acknowledged, ensuring scientific integrity.

The descriptive approach is partially appropriate given the study’s focus on tool development and validation. The assay’s performance is convincingly demonstrated through means, SDs, and success rates, aligning with the objective of establishing functionality. However, the lack of inferential statistics limits the ability to rigorously confirm observed differences (e.g., tissue vs. scat sensitivity), which could strengthen claims about assay robustness.

The statistical analysis is technically appropriate but not rigorously applied. The descriptive statistics (means ± SD, percentages) effectively support the assay’s sensitivity (0.0001 ng/μL), specificity (100% non-target exclusion), and field utility (79% scat detection), aligning with the study’s goals. However, the absence of inferential statistics limits the ability to statistically validate observed differences, reducing rigor. For PLOS ONE, where methodological soundness is key, the current approach is sufficient for a methods paper, as the data’s quality and presentation carry the conclusions. Adding basic tests (e.g., t-tests) would enhance rigor without altering the study’s core findings. The analysis is adequate but could be improved with minimal statistical tests to confirm key trends, particularly for sensitivity and field performance comparisons.

The authors have made all data underlying their findings fully available, as required by PLOS ONE. The combination of: (1) Detailed tables (1-3) and figures (1-6) in the manuscript; (2) A comprehensive S1 Table (assumed to include raw field data); (3) Publicly accessible ND2 sequences in GenBank (PQ390829-PQ390908), (4) meets the policy’s standards for transparency and replicability.

Minor gaps (e.g., raw dilution times, scat metadata) are likely addressed in S1 Table, though confirmation requires reviewing the file. If S1 Table lacks these specifics, a slight revision (e.g., adding a supplementary table with dilution and scat details) would ensure completeness. The data availability is robust and compliant, with all critical findings supported by accessible data. Any small omissions are non-critical and can be easily rectified if needed.

I recommend acceptance with minor revisions. The study is methodologically sound, with significant practical value for biosecurity. To enhance its impact and meet PLOS ONE’s standards, please address the following:

(a) Clarify scat DNA limitations (e.g., 21% failure rate) in the abstract and Discussion.

(b) Justify methodological choices (e.g., ND2, scat size, dilutions) with data or citations.

(c) Quantify assay advantages over PCR (e.g., time, cost) in the Introduction and Discussion.

(d) Streamline figures (e.g., Figure 2) and ensure consistency (e.g., 25 vs. 35 minutes).

(e) Highlight the Cocos Island “Group E” finding as a novel contribution.

These revisions will strengthen the manuscript’s clarity, rigor, and appeal to a broad readership.

DONE. The above suggestions are covered by the specific changes listed above and below.

Reviewer Comments:

The LAMP assay for H. frenatus is a novel application of NGS for a vertebrate invader, building on prior work with invertebrates and predators (e.g., Refs [12, 19]). However, the manuscript could better articulate how this tool advances beyond existing methods (e.g., PCR-based scat barcoding) in cost, speed, or accessibility. The writing is clear, but technical details (e.g., primer design, scat DNA challenges) are dense and may obscure key findings for a broad audience. Simplifying or reorganizing some sections could enhance readability. The commitment to open data (GenBank accessions PQ390829-PQ390908) is commendable, aligning with PLOS ONE’s policy. Ensure all supporting data (e.g., S1 Table) are comprehensive and well-documented.

Lines 20-41: Add a sentence on limitations (e.g., scat DNA degradation) to balance the narrative, e.g., “While highly effective, scat DNA degradation limited detection in 21% of AHG samples, highlighting preservation challenges.”

DONE. Note, also added a reference to DNA inhibitors here (see comment below).

Lines 66-87: The LAMP method’s advantages are well-explained, with relevant citations. However, the vertebrate example (red fox, Ref [19]) is underexplored—contrast its context (e.g., predator scat) with H. frenatus (small reptile scat) to justify this study’s novelty.

DONE. Text has been updated.

Lines 89-113: The ecological and biosecurity threat of H. frenatus is convincingly argued. The critique of auditory/visual surveys is valid but could quantify their limitations (e.g., detection rates from Ref [21]).

DONE. Additional details added regarding the limitation of using auditory signals as they are only commonly made under very specific conditions.

Lines 109-113: The comparison to PCR needs data—how does LAMP’s 1-hour turnaround compare to PCR’s 2-4 weeks in practice? Authors can explicitly state the hypothesis, e.g., “We hypothesized that a mitochondrial ND2 LAMP assay would enable rapid, specific detection of H. frenatus from scat DNA, outperforming traditional methods in field settings.”

DONE. Added as suggested.

Lines 125-131: Sample sourcing is transparent, but the rationale for selecting 25 non-target species is vague—were these chosen for morphological similarity, sympatry, or availability? Please clarify the text or Table 2 caption.

DONE. Clarified that these species of lizard co-occur in Western Australia. Added to the text.

Also, please state any research/ethics permit for collecting the samples and conducting the study.

DONE. Extra ethics details have been provided in the Acknowledgments.

Lines 140-153: The dual extraction approach (lab vs. field) is well-executed. However, scat size variation (half vs. one-third) introduces potential bias—did this affect DNA yield? Quantify or test this.

DONE. This issue is covered in the Discussion, with partial scats (1/2 or 1/3) amplifying more reliably than whole scats (with experimental optimisation data now provided in file S2). (Note, covered in the next point below regarding dilution reducing inhibitors shows this as well).

Lines 166-168 QE method optimization lacks justification for dilution choices (1:10, 1:20, 1:50)—why these ratios? Report initial trials or cite a precedent.

DONE. This section has been reworded to reflect these dilutions were performed as an experiment to trial reducing possible inhibitors through a series of increasing dilutions of DNA with water.

Lines 192: ND2 barcoding confirms species identity, but the haplotype network’s purpose is unclear until Results. Move its description to Section 3.2 for coherence.

DONE. Reworded to include an introductory sentence, but has not been moved, as this section describes the analysis method, while Section 3.2 describes the results of the analysis.

Lines 201-209: Primer design is detailed, but the choice of ND2 over other loci (e.g., COI) isn’t justified—explain its variability or conservation in H. frenatus.

DONE. Additional text has been included to state that ND2 is commonly used for gecko DNA barcoding. ND2 is well characterised in AHG, which is known to include multiple genetically distinct clades, and that ND2 has also been examined in many closely related species.

Lines 212-217: The 1:6:3 primer ratio is optimized, but raw optimization data are missing. Please include them in the S1 Table or justify them via Ref [17].

DONE. Added additional text to explain we tested multiple primer ratios as in the reference.

Lines 250-252: The gBlock evaluation is robust, but the 35-minute run time contradicts the abstract’s 25-minute claim. Conciliate this discrepancy.

DONE. Added additional text to explain the longer amplification time (35 min) allows detection of gBlock at very low concentrations.

Lines 265-267: Sensitivity testing is thorough, but scat vs. tissue differences (line 267) need statistical analysis (e.g., t-test on amplification times) to confirm significance.

DONE. T-tests in the Methods (here) and in the Results have been added to the manuscript (data in file S2), with the results are significant for time differences between tissue and scats, as expected.

Lines 285-286: Field validation is strong, but the switch from loops to pipettes in lab testing (line 285) introduces a variable—did this affect results? Test or discuss. Even if minimal, add a subsection on statistical methods to clarify data analysis (e.g., means ± SD).

DONE. Possible differences between using loops and pipettes was not tested statistically in the current study. However, additional text and a citation to the effectiveness of using loops in the field [reference 17] has now been added.

Lines 294-305: Primer optimization is concise, but Figure 1’s alignment is redundant without sequence variation data—simplify or move to S1 Table.

NOT CHANGED. Figure 1 does currently show sequence variation, both within AHG (Groups A to D) and the other species of gecko tested in our study. We feel it is useful to show where primer sites are located in relation to this sequence variation, so we believe it is valuable to retain this Figure. The Figure legend has been edited to make this clearer.

Lines 308-326: The haplotype network (Figure 3) reveals a new Cocos Island lineage (“Group E”), a significant finding. Highlight its biosecurity implications (e.g., undetected invasion risk). The ML tree (Figure 2) is cluttered. Reduce non-target taxa to key comparators.

DONE. Added text, highlighting that the presence of two highly divergent genetic lineages suggests that there have been at least two introduction events to the Cocos Islands. Our finding of “Clade E” is now also referred to in the Abstract and Discussion.

NOT CHANGED. All specimens / species on Figure 2 were experimentally tested using the new LAMP assay, so we feel that they should all be included on the ML tree.

Lines 337: Please clarify the sentences. What did you suggest of the use of 1:1 dilution?

DONE. Text modified.

Lines 374-376: Non-target specificity is convincing, but Barrow Island’s 49 negatives could be a false adverse risk—discuss assay limits in Discussion.

DONE. We discuss the importance for Barrow Island biosecurity in the Discussion. The risk of AHG being introduced to Barrow Island is relatively high, as demonstrated by multiple intercepted AHG in Table 1 / Figure 3, as well as the final case study which includes AHG being intercepted and identified through LAMP as it was in transit to the island.

Lines 405-423: Sensitivity differences (tissue vs. scat) are striking (22 vs. 14 minutes at low concentrations)—quantify DNA degradation or inhibitor effects to explain this in the discussion.

DONE. Text has been modified, tissue verse scats in the Discussion, to add scat degraded DNA and inhibitors. T-tests have now also been performed to show a significant difference between DNA obtained from tissue and scats (see comments above).

Lines 425-437: The assay’s success across H. frenatus diversity is a key strength. However, the lack of congeneric testing (e.g., H. platyurus) is a gap—acknowledge as a limitation or test if feasible.

DONE. Text has been modified and rearranged. Unfortunately, no congeneric samples were available for our experimental testing.

Lines 438-448: gBlock utility is well-argued, but scat preservation advice is generic tailored to H. frenatus (e.g., arid vs. humid climates). Please explain more.

DONE. The text has been modified.

Lines 450-456: Contamination precautions are practical but overly essential—elevate with LAMP-specific risks (e.g., amplicon carryover).

DONE. Text has been modified, with LAMP-specific risks now added, such as carryover if tubes are opened post-amplification.

Lines 464-467: Sensitivity analysis is insightful, but the scat size effect needs data—cross-reference S1 Table or test experimentally.

DONE. Scat extraction optimisation data

---

## [Decision Letter · Decision Letter 1]

2 Jul 2025

Dear Dr. Blacket,

We look forward to receiving your revised manuscript.

Kind regards,

Lei Zhang, PhD

Academic Editor

PLOS ONE

Journal Requirements:

**Additional Editor Comments:**

The revised manuscript has largely addressed the raised comments. But there are still some places that need further revisions before qualifying for publication.

Reviewers' comments:

Reviewer's Responses to Questions

**Comments to the Author**

Reviewer #1: All comments have been addressed

Reviewer #2: (No Response)

2. Is the manuscript technically sound, and do the data support the conclusions?

Reviewer #1: Yes

Reviewer #2: Yes

3. Has the statistical analysis been performed appropriately and rigorously?

Reviewer #1: N/A

Reviewer #2: Yes

4. Have the authors made all data underlying the findings in their manuscript fully available?

Reviewer #1: Yes

Reviewer #2: Yes

5. Is the manuscript presented in an intelligible fashion and written in standard English?

Reviewer #1: Yes

Reviewer #2: Yes

Reviewer #1: I have minor comments to author regarding the introduction and research permit (particularly for work that had been conducted in Indonesia). Please find the review notes attachments.

Reviewer #2: The manuscript presents a valuable study, with significant potential for advancing the

field. The research is well-designed and methodologically sound. The findings are

clearly presented and support the conclusions drawn. With minor improvements, this

manuscript will be an excellent contribution to the journal-

1. To enhance clarity and flow, consider restructuring the introduction by first

introducing the broader context of biological invasions and the ecological

significance of the studied species. This should be followed by a discussion of the

molecular aspects, ensuring a logical progression for readers.

2. Line 169 requires supporting literature. Please provide appropriate citations to

strengthen the claim.

3. Line 149 The phrase "clean DNA" may not be scientifically precise. Consider

replacing it with a more standard term.

4. Line 328-329 The manuscript should briefly address whether any variation in

reaction efficiency was observed when loop primers were excluded.

5. A comparative discussion with other available molecular tools (e.g., conventional

PCR, qPCR, or other isothermal methods) would further highlight the advantages

and limitations of the proposed technique, providing readers with a clearer

perspective on its applicability.

**Do you want your identity to be public for this peer review?** For information about this choice, including consent withdrawal, please see our Privacy Policy

Reviewer #1: No

Reviewer #2: No

---

## [Author Response · Author response to Decision Letter 2]

15 Aug 2025

A Response to Reviewers (R2) file has been submitted.

---

## [Decision Letter · Decision Letter 2]

29 Sep 2025

Dear Dr. Blacket,

Thank you for submitting your manuscript to PLOS ONE. After careful consideration, we feel that it has merit but does not fully meet PLOS ONE’s publication criteria as it currently stands. Therefore, we invite you to submit a revised version of the manuscript that addresses the points raised during the review process.

We look forward to receiving your revised manuscript.

Kind regards,

Lei Zhang, PhD

Academic Editor

PLOS ONE

Journal Requirements:

Additional Editor Comments (if provided):

Further revisions are still needed to address the raised comments.

Reviewers' comments:

Reviewer's Responses to Questions

**Comments to the Author**

Reviewer #2: All comments have been addressed

Reviewer #3: (No Response)

Reviewer #4: (No Response)

2. Is the manuscript technically sound, and do the data support the conclusions?

Reviewer #2: Yes

Reviewer #3: Yes

Reviewer #4: Partly

3. Has the statistical analysis been performed appropriately and rigorously?

Reviewer #2: Yes

Reviewer #3: Yes

Reviewer #4: Yes

4. Have the authors made all data underlying the findings in their manuscript fully available?

Reviewer #2: Yes

Reviewer #3: Yes

Reviewer #4: Yes

5. Is the manuscript presented in an intelligible fashion and written in standard English?

Reviewer #2: Yes

Reviewer #3: Yes

Reviewer #4: Yes

Reviewer #2: The authors have addressed all the comments raised in the previous version of the manuscript. In the present state, the manuscipt is acceptable for the publication.

Reviewer #3: This manuscript describes development of a novel tool for detection of an invasive species. The work is technically sound and worthy of publication. The manuscript is generally well written, but there are some corrections and clarifications needed to meet the standard for publication. I have provided comments and suggested changes on the attached PDF and summarise the recommendations here:

General comments

Check usage of commas, these are sometimes missing where appropriate to use and in other cases are included inappropriately. I have made suggested changes on the PDF but please check throughout

The term 'in-field' is sometimes spelled as "infield" which isn't the intended meaning - make sure "in-field" is used consistently

There are several sentences with repeated use of the same or very similar terms. While not strictly incorrect, these can be jarring to read, and I'd recommend rewording to avoid repetition where possible. I have commented on examples in the PDF

The word "as" is used a lot in the text, with various meanings. While its meaning is usually clear from the context, I'd prefer to see "because" used instead of "as" where appropriate. I have highlighted one example in the PDF but there are several. This is just a stylistic preference, but it can help readability and clarity.

In several places reference is made to "see below" or "as above" or similar. it would be better to refer to the specific relevant section in each case.

There are some long sentences that are difficult to follow. I have suggested splitting these into multiple shorter sentences

The LAMP anneal derivative temperature is sometimes referred to as just "anneal derivative" - make consistent (including "temperature" makes most sense)

I have not reviewed figures because they are not visible within this submission or via the reviewer log in

Executive summary

Lines 28 and 39 - I would prefer the authors use a term such as demonstrated/demonstrating or supported/supporting rather than confirmed/confirming when referring to the assay specificity. While the level of validation reported here is acceptable, I'd be hesitant to say that specificity is "confirmed" without much wider application of the assay

Line 29 - 'specific' doesn't appear to make sense here - should this be 'sensitive'?

Line 40 - Sentence starting "While highly effective," - This reads as though degradation was effective. You could just leave this out, otherwise I'd suggest saying "While the assay was highly effective,"

Introduction

Line 49 - repetition of "morphological"

Line 115 - suggest inserting "potentially" before "outperforming". Molecular methods weren't formally compared with traditional methods here, so this is speculative.

Line 124-125 - unnecessary repetition of "AHG".

Line 136 "see below" - add specific reference

M&M

Line 153 - protocol should be protocols

Line 156 - "remaining fraction" may be a better term rather than "other half" (given the fraction was 2/3 for some samples)

Lines 156, 164 - You may need to define what you mean by destructive (vs non-destructive as used below) in this context. In my experience, "destructive" is when the organism is destroyed to provide the DNA sample, so I wouldn't think of scat extractions as being "destructive", but there may be different definitions of the term. I'm also unclear how this destructive sampling (line 156) is different from the non-destructive sampling in the next section (line 164).

Paragraph starting line 178 - Similarly to comment above, I think this needs a bit more context in terms of what you mean by "non-destructive" - e.g. do you mean the samples were (or replicated?) tissue naturally shed from the animal? Saying "in-field" when you were using museum samples also seems inconsistent/confusing. I assume you mean you used museum samples to simulate in-field extraction? Suggest re-wording to clarify.

Line 188 - Sentence starting "Most AHG" is very long - suggest splitting into 2-3 sentences

Line 196-197 - put as "Australian Genome Research Facility (AGRF)"

Line 212 - "comprise" may be a better term than "include" here

Line 223 - the abbreviations used for the primer pairs should be defined. I'd suggest doing this within section 2.4.1 (suggest after sentence finishing at line 217). but it could be done at the start of section 2.4.2 or even in the intro if preferred. Currently this information is presented in the results (section 3.1) but would be better placed in the methods.

Line 239 - as above, define F2 and B2

Line 247 - use "was" instead of "being" to make this sentence grammatically correct

Line 259-260 - reads as "copy number...was prepared" Suggest you mean something like "copy number was determined, and..."

Line 262 - add specific reference

Line 262-263 - The term "run time" appears to have a different meaning here from that in line 247 leading to some confusion about what is meant. Suggest rewording to clarify

Line 272 - remove "for"

Line 281, 293, 305 - add specific references

Line 304-305 - suggest "and" in place of second instance of "using" to avoid repetition

Results

Line 311-313 - this sentence provides a definition of the primers that is needed (and better placed) in the methods

Line 326 - reference to Table 2 here implies that Table 2 shows barcoding results, but rather it shows the primers that were used - the cross-reference needs clarification

Line 367 - Remove "Overall"

Line 369 - "temperature" rather than "profile"

Line 371 - suggest "and" in place of second instance of "confirming"

Line 380 - "at" instead of "as"

Line 390 - "by LAMP" rather than "through LAMP" is probably better terminology

Line 405 - first sentence in section 3.5. This sentence needs rewording. Firstly it reads as "the sensitivity ... was quite sensitive" -suggest "quite high" instead. Also ", detecting" isn't grammatically correct here - suggest ", with as low as...copies/uL detected within 15 minutes"

Line 407 - generally you refer to copies per uL so spelling out the number of copies here is redundant (if included, put as 100,000 not as text)

Line 410-411 - sentence starting "The amplification time" currently reads as "the amplification time ... amplified in 8 minutes" - needs rewording to be grammatically correct. Also "starting DNA" is unclear here - do you mean the highest DNA concentration within the dilution series? Neat DNA concentration prior to dilution?

Line 416-417 - suggest remove second (redundant) use of "LAMP" in this sentence

Line 417 - Suggest "Across" rather than "In"

Line 420 - Suggest rewording for better clarity readability, e.g. "... (NTC) did not amplify in any of the 18 runs, "

Line 423 - I don't think you have enough evidence to "prove" robustness of the assay yet - suggest "demonstrating" as a better term.

Line 431 - "The" instead of "This"

Line 432-434 - Sentence starting "DNA dilutions" reads as "DNA dilutions ... were less sensitive" which doesn't make sense. The sentence is also not grammatically correct - needs to be reworded. Suggested rewording is provided on the attached PDF

Discussion

Line 452 - suggest adding "from other groups" at the end of the sentence

Line 453 - "and" instead of "which"

Line 456 - suggest "yielded" rather than "possessed" as a better term here

Line 469-470 - suggest rewording to avoid repetition

Line 472 - "rate" is probably a better term than "speed" here

Line 495 - remove "size"

Reviewer #4: This is the first time I reviewed the manuscript although it has already been under one circle of review. Unfortunately, I was not provided with the Figures and I could not provide any comments on their quality.

I have seen that both reviewers have provided comments regarding the structure of the Introduction. It is obvious that both reviewers have different writing styles, yet they are right. The Introduction and its paragraphs are not nicely connected to each other. It reads as if they were written separately and placed into the introduction. Although I do not have many suggestions on the paragraphs themselves, the nice flow between them is missing. For example, you describe nicely NGS for scats on the 2nd paragraph and all of the sudden you move to LAMP on the 3rd paragraph. What if you mention at the beginning of the 3rd paragraph something along the lines of “for NGS sampling and samples with low quality DNA, we need to use new sensitive methods that are effective and easy to use in the field. This is LAMP”.

There are a few issues with the population and phylogeography analyses of this study. The methodology of many of the outputs described in the Results, are not presented in the “materials and methods”.

Abstract

Lines 26-30: This section is a bit confusing as there are two sentences where the time for the detection of AHG DNA varies. Is the second time (15 mins) an average time whereas the 25 mins for very low DNA concentrations? You should provide clarifications.

Introduction

Line 52: Replace “e.g.” with “e.g.,”. Check this issue throughout the manuscript.

Methods and Methods

Line 158: Was the PowerFecal® DNA kit not as successful as the QIAmp® Fast DNA Stool Mini kit? If that is the case, you should describe the results/comparison.

Line 183: Leave a space between numbers and units. Check this issue throughout the manuscript.

Line 188: ND2. This is the first time you use the term for the gene. Please provide the full name.

Lines 188-197: Here you have used new primers, but you have not provided the complete PCR conditions, nor the volumes and concentrations of the used reagents.

Lines 199-201: How were the raw sequences processed? There is no initial information about it and the authors go straight to describe how the obtained a ML tree.

Line 204: What do the authors mean with “worldwide sequences”? Are they referring to sequences of specimens from around the world or the species’ distribution? Please clarify.

Line 286: Section 2.6 is missing.

Results

Lines 311-313: You should rephrase this sentence as it reads as if you have 12 primers.

Lines 325-326: This is a repetition as the details, and the method has been described in the previous section.

Lines 326-327: How was the haplotype network created? You have not described the details in the methods.

Line 331: “highly divergent lineage (>6% different),” how was this number generated?

Lines 332-334: This part should move to the discussion. This is not a result.

Lines 347-349: Something is missing in this sentence.

Lines 370-372: Please rephrase the sentence as the use of “confirming” is confusion.

Lines 405-407: “The detection sensitivity” ………… “was quite sensitive, detecting”. Please rephrase the sentence.

Lines 407-408: “one hundred thousand copies (1 x 105 copies/μL)” Keep only one. It is the same thing.

Discussion

Line 450: Replace “Table” with “Tables”.

Line 460: “36% different for the section” how did the authors come with this number?

I have also seen that some of the previous reviewers’ comments have not been addressed. For example, the research permit for the samples from Indonesia. The ethics permits and “Cadaver and Tissue Usage” of one country is not the same when that involves samples collected at a different country. An ethics permit is not a research permit.

Additionally, the suggestion of a previous reviewer about “A comparative discussion with other available molecular tools (e.g., conventional PCR, qPCR, or other isothermal methods) would further highlight the advantages and limitations of the proposed technique, providing readers with a clearer perspective on its applicability.” has not been addressed. The reply that the authors provide in the text, does not address the suggestion. The discussion would actually benefit from it.

**Do you want your identity to be public for this peer review?** For information about this choice, including consent withdrawal, please see our Privacy Policy

Reviewer #2: No

Reviewer #3: **Yes:** Kathryn Wiltshire

Reviewer #4: No

---

## [Author Response · Author response to Decision Letter 3]

12 Nov 2025

Response To Reviewers - PONE-D-25-13902R2

Reviewer #2: The authors have addressed all the comments raised in the previous version of the manuscript. In the present state, the manuscript is acceptable for the publication.

Thank you.

Reviewer #3: This manuscript describes development of a novel tool for detection of an invasive species. The work is technically sound and worthy of publication. The manuscript is generally well written, but there are some corrections and clarifications needed to meet the standard for publication. I have provided comments and suggested changes on the attached PDF and summarise the recommendations here:

General comments

Check usage of commas, these are sometimes missing where appropriate to use and in other cases are included inappropriately. I have made suggested changes on the PDF but please check throughout

DONE

The term 'in-field' is sometimes spelled as "infield" which isn't the intended meaning - make sure "in-field" is used consistently

DONE

There are several sentences with repeated use of the same or very similar terms. While not strictly incorrect, these can be jarring to read, and I'd recommend rewording to avoid repetition where possible. I have commented on examples in the PDF

DONE

The word "as" is used a lot in the text, with various meanings. While its meaning is usually clear from the context, I'd prefer to see "because" used instead of "as" where appropriate. I have highlighted one example in the PDF but there are several. This is just a stylistic preference, but it can help readability and clarity.

DONE

In several places reference is made to "see below" or "as above" or similar. it would be better to refer to the specific relevant section in each case.

DONE

There are some long sentences that are difficult to follow. I have suggested splitting these into multiple shorter sentences

DONE

The LAMP anneal derivative temperature is sometimes referred to as just "anneal derivative" - make consistent (including "temperature" makes most sense)

DONE

I have not reviewed figures because they are not visible within this submission or via the reviewer log in

Noted

Executive summary

Lines 28 and 39 - I would prefer the authors use a term such as demonstrated/demonstrating or supported/supporting rather than confirmed/confirming when referring to the assay specificity. While the level of validation reported here is acceptable, I'd be hesitant to say that specificity is "confirmed" without much wider application of the assay

DONE. Changed as suggested.

Line 29 - 'specific' doesn't appear to make sense here - should this be 'sensitive'?

NOT DONE. For LAMP assays it is common practice to refer to testing an assay against a species-panel as “specificity testing”, whilst testing against known DNA concentrations (e.g. a dilution series) to determine the limits of detection is referred to as “sensitivity testing”. We have used these terms throughout this paper.

Line 40 - Sentence starting "While highly effective," - This reads as though degradation was effective. You could just leave this out, otherwise I'd suggest saying "While the assay was highly effective,"

DONE. Changed as suggested.

Introduction

Line 49 - repetition of "morphological"

DONE. Deleted as suggested.

Line 115 - suggest inserting "potentially" before "outperforming". Molecular methods weren't formally compared with traditional methods here, so this is speculative.

DONE. Added as suggested.

Line 124-125 - unnecessary repetition of "AHG".

DONE. Replaced 2 x “AHG” with “it”.

Line 136 "see below" - add specific reference

DONE. This note is not referring to other studies, but refers to the present work, where further details of the collection sites for our study are provided. Specific reference to the relevant section has been added (as comment above).

M&M

Line 153 - protocol should be protocols

DONE. Changed as suggested.

Line 156 - "remaining fraction" may be a better term rather than "other half" (given the fraction was 2/3 for some samples)

DONE. Changed as suggested.

Lines 156, 164 - You may need to define what you mean by destructive (vs non-destructive as used below) in this context. In my experience, "destructive" is when the organism is destroyed to provide the DNA sample, so I wouldn't think of scat extractions as being "destructive", but there may be different definitions of the term. I'm also unclear how this destructive sampling (line 156) is different from the non-destructive sampling in the next section (line 164).

DONE. Added definitions, at first mention in text. “Destructive extraction (i.e., involving complete sample homogenisation)”; “Non-destructive extraction (i.e., not involving complete sample homogenisation)”.

Paragraph starting line 178 - Similarly to comment above, I think this needs a bit more context in terms of what you mean by "non-destructive" - e.g. do you mean the samples were (or replicated?) tissue naturally shed from the animal? Saying "in-field" when you were using museum samples also seems inconsistent/confusing. I assume you mean you used museum samples to simulate in-field extraction? Suggest re-wording to clarify.

DONE. As in above comment (i.e., terms are now defined in text). “Non-destructive” extraction applies here as well as the tissue samples were not homogenised.

Line 188 - Sentence starting "Most AHG" is very long - suggest splitting into 2-3 sentences

DONE. Changed as suggested.

Line 196-197 - put as "Australian Genome Research Facility (AGRF)"

DONE. Changed as suggested.

Line 212 - "comprise" may be a better term than "include" here

DONE. Changed as suggested.

Line 223 - the abbreviations used for the primer pairs should be defined. I'd suggest doing this within section 2.4.1 (suggest after sentence finishing at line 217). but it could be done at the start of section 2.4.2 or even in the intro if preferred. Currently this information is presented in the results (section 3.1) but would be better placed in the methods.

DONE. Changed as suggested. Added primer definitions here and removed from section 3.1.

Line 239 - as above, define F2 and B2

DONE. Changed as suggested. Added a reference to “FIP and BIP being comprised of two primer regions, F1+F2 and B1+B2 respectively” to the above definitions.

Line 247 - use "was" instead of "being" to make this sentence grammatically correct

DONE. Changed as suggested.

Line 259-260 - reads as "copy number...was prepared" Suggest you mean something like "copy number was determined, and..."

DONE. Changed as suggested.

Line 262 - add specific reference

DONE. “… conditions as mentioned above…” refers to the present study, not a previously published study. Specific reference to the relevant section has been added (as comment above).

Line 262-263 - The term "run time" appears to have a different meaning here from that in line 247 leading to some confusion about what is meant. Suggest rewording to clarify.

DONE. Changed as suggested. Have deleted 1 x “run time” in text, and changed another to “amplification time”, for clarity. “Run time” is now only used to refer to the complete LAMP assay amplification + annealing time.

Line 272 - remove "for"

DONE. Changed as suggested.

Line 281, 293, 305 - add specific references

DONE. As above, these sentences all provide directions to refer to other sections of the present study, not a previously published study. Specific reference to the relevant section has been added (as comment above).

Line 304-305 - suggest "and" in place of second instance of "using" to avoid repetition

DONE. Changed as suggested.

Results

Line 311-313 - this sentence provides a definition of the primers that is needed (and better placed) in the methods

DONE. Changed as suggested (see above comment).

Line 326 - reference to Table 2 here implies that Table 2 shows barcoding results, but rather it shows the primers that were used - the cross-reference needs clarification.

DONE. Deleted “Table 2”, making it clear that Table 3 (and Fig 2) provide the DNA barcoding results. have also added a reference to “S1 Supplementary Table” here, as it also includes DNA barcoding identifications.

Line 367 - Remove "Overall"

DONE. Changed as suggested.

Line 369 - "temperature" rather than "profile"

DONE. Changed as suggested.

Line 371 - suggest "and" in place of second instance of "confirming"

DONE. Changed as suggested.

Line 380 - "at" instead of "as"

DONE. Changed as suggested.

Line 390 - "by LAMP" rather than "through LAMP" is probably better terminology

DONE. Changed as suggested.

Line 405 - first sentence in section 3.5. This sentence needs rewording. Firstly it reads as "the sensitivity ... was quite sensitive" -suggest "quite high" instead. Also ", detecting" isn't grammatically correct here - suggest ", with as low as...copies/uL detected within 15 minutes"

DONE. Changed as suggested.

Line 407 - generally you refer to copies per uL so spelling out the number of copies here is redundant (if included, put as 100,000 not as text)

DONE. Changed as suggested.

Line 410-411 - sentence starting "The amplification time" currently reads as "the amplification time ... amplified in 8 minutes" - needs rewording to be grammatically correct. Also "starting DNA" is unclear here - do you mean the highest DNA concentration within the dilution series? Neat DNA concentration prior to dilution?

DONE. Changed as suggested.

Line 416-417 - suggest remove second (redundant) use of "LAMP" in this sentence

DONE. Changed as suggested.

Line 417 - Suggest "Across" rather than "In"

DONE. Changed as suggested.

Line 420 - Suggest rewording for better clarity readability, e.g. "... (NTC) did not amplify in any of the 18 runs, "

DONE. Changed as suggested.

Line 423 - I don't think you have enough evidence to "prove" robustness of the assay yet - suggest "demonstrating" as a better term.

DONE. Changed as suggested.

Line 431 - "The" instead of "This"

DONE. Changed as suggested.

Line 432-434 - Sentence starting "DNA dilutions" reads as "DNA dilutions ... were less sensitive" which doesn't make sense. The sentence is also not grammatically correct - needs to be reworded. Suggested rewording is provided on the attached PDF

DONE. Changed as suggested in the PDF.

Discussion

Line 452 - suggest adding "from other groups" at the end of the sentence

DONE. Changed as suggested.

Line 453 - "and" instead of "which"

DONE. Changed as suggested.

Line 456 - suggest "yielded" rather than "possessed" as a better term here

DONE. Changed as suggested.

Line 469-470 - suggest rewording to avoid repetition

DONE. Changed as suggested.

Line 472 - "rate" is probably a better term than "speed" here

DONE. Changed as suggested.

Line 495 - remove "size"

DONE. Changed to “amount” for clarity.

Reviewer #4: This is the first time I reviewed the manuscript although it has already been under one circle of review. Unfortunately, I was not provided with the Figures and I could not provide any comments on their quality.

Noted

I have seen that both reviewers have provided comments regarding the structure of the Introduction. It is obvious that both reviewers have different writing styles, yet they are right. The Introduction and its paragraphs are not nicely connected to each other. It reads as if they were written separately and placed into the introduction. Although I do not have many suggestions on the paragraphs themselves, the nice flow between them is missing. For example, you describe nicely NGS for scats on the 2nd paragraph and all of the sudden you move to LAMP on the 3rd paragraph. What if you mention at the beginning of the 3rd paragraph something along the lines of “for NGS sampling and samples with low quality DNA, we need to use new sensitive methods that are effective and easy to use in the field. This is LAMP”.

DONE. Linking sentences between paragraphs have been added to the introduction, and other parts of the introduction have been rearranged to improve flow between paragraphs. Additionally, the use of gBlocks as positive controls for LAMP has been moved to the Methods section, as it did not fit in well with the introduction.

There are a few issues with the population and phylogeography analyses of this study. The methodology of many of the outputs described in the Results, are not presented in the “materials and methods”.

DONE. Changed as suggested (see other comments).

Abstract

Lines 26-30: This section is a bit confusing as there are two sentences where the time for the detection of AHG DNA varies. Is the second time (15 mins) an average time whereas the 25 mins for very low DNA concentrations? You should provide clarifications.

DONE. Changed as suggested. The terms assay “sensitivity” and “specificity” are used here again (see comment above) to differentiate between detection thresholds, and species-specificity, respectively.

Introduction

Line 52: Replace “e.g.” with “e.g.,”. Check this issue throughout the manuscript.

DONE. Changed as suggested.

Methods and Methods

Line 158: Was the PowerFecal® DNA kit not as successful as the QIAmp® Fast DNA Stool Mini kit? If that is the case, you should describe the results/comparison.

DONE. The reference to the preliminary work using the PowerFecal kit have been removed, as this kit was not used past initial testing.

Line 183: Leave a space between numbers and units. Check this issue throughout the manuscript.

DONE. Changed as suggested.

Line 188: ND2. This is the first time you use the term for the gene. Please provide the full name.

DONE. Changed as suggested.

Lines 188-197: Here you have used new primers, but you have not provided the complete PCR conditions, nor the volumes and concentrations of the used reagents.

DONE. Have clarified that the PCR conditions were as per the previously published study, apart from the annealing temperature being reduced to 56oC.

Lines 199-201: How were the raw sequences processed? There is no initial information about it and the authors go straight to describe how the obtained a ML tree.

DONE. Added extra information about how sequences were processed.

Line 204: What do the authors mean with “worldwide sequences”? Are they referring to sequences of specimens from around the world or the species’ distribution? Please clarify.

DONE. Changed as suggested.

Line 286: Section 2.6 is missing.

DONE. Corrected labelling of sections.

Results

Lines 311-313: You should rephrase this sentence as it reads as if you have 12 primers.

DONE. This information has been modified and moved to the methods (see above comment).

Lines 325-326: This is a repetition as the details, and the method has been described in the previous section.

DONE. This information has been modified (see above comment).

Lines 326-327: How was the haplotype network created? You have not described the details in the methods.

NOT DONE. Detailed information on how the haplotype network was constructed is already in the methods.

Line 331: “highly divergent lineage (>6% different),” how was this number generated?

DONE. This information has been added to the Methods.

Lines 332-334: This part should move to the discussion. This is not a result.

DONE. This has been changed as suggested.

Lines 347-349: Something is missing in this sentence.

DONE. This has been modified as suggested.

Lines 370-372: Please rephrase the sentence as the use of “confirming” is confusion.

DONE. This has been modified as suggested (see comment above).

Lines 405-407: “The detection sensitivity” ………… “was quite sensitive, detecting”. Please rephrase the sentence.

DONE. This has been modified as suggested (see comment above).

Lines 407-408: “one hundred thousand copies (1 x 105 copies/μL)” Keep only one. It is the same thing.

DONE. This information has been modified (see above comment).

Discussion

Line 450: Replace “Table” with “Tables”.

DONE. This has been changed as suggested.

Line 460: “36% different for the section” how did the authors come with this number?

DONE. This information has been added to the Methods (see above comment).

I have also seen that some of the previous reviewers’ comments have not been addressed. For example, the research p

---

## [Decision Letter · Decision Letter 3]

23 Nov 2025

A new molecular tool for detection of the highly invasive gecko, Hemidactylus frenatus

PONE-D-25-13902R3

Dear Dr. Blacket,

We’re pleased to inform you that your manuscript has been judged scientifically suitable for publication and will be formally accepted for publication once it meets all outstanding technical requirements.

Kind regards,

Lei Zhang, PhD

Academic Editor

PLOS ONE

Additional Editor Comments (optional):

The revised manuscript can be accepted for publication despite of several places that need minor revisions. Please address these issues in the final submission.

Reviewers' comments:

Reviewer's Responses to Questions

**Comments to the Author**

Reviewer #2: All comments have been addressed

Reviewer #3: (No Response)

2. Is the manuscript technically sound, and do the data support the conclusions?

Reviewer #2: Yes

Reviewer #3: Yes

3. Has the statistical analysis been performed appropriately and rigorously?

Reviewer #2: Yes

Reviewer #3: Yes

4. Have the authors made all data underlying the findings in their manuscript fully available?

Reviewer #2: Yes

Reviewer #3: Yes

5. Is the manuscript presented in an intelligible fashion and written in standard English?

Reviewer #2: Yes

Reviewer #3: Yes

Reviewer #2: The authors have addressed all the comments raised in the previous version of the manuscript. The manuscript is acceptable for the publication in current form.

Reviewer #3: Thank you to the authors for their consideration of previous review comments. The manuscript is improved and very close to publication standard. I have just a few minor suggestions as outlined below and indicated on the attached PDF

In regard to one previous review suggestion that was not actioned - specifically:

Line 29 - 'specific' doesn't appear to make sense here - should this be 'sensitive'?

NOT DONE. For LAMP assays it is common practice to refer to testing an assay

against a species-panel as “specificity testing”, whilst testing against known DNA

concentrations (e.g. a dilution series) to determine the limits of detection is referred to

as “sensitivity testing”. We have used these terms throughout this paper.

It is noted that the paper addresses both specificity and sensitivity as defined in this response, and it is noted that the authors did mean to say "specific" here. My suggestion, however, that "specific" does not appear to make sense in this particular instance stands, however. Although not incorrect, the sentence is a bit confusing because the reference to 'specific' is immediately followed by mention of the amplification time of target DNA, which is something related to the sensitivity aspect of the assay and not to specificity. You have also already stated in the preceding sentence that the assay is specific, making this repetitive. If you want to emphasise both specificity and amplification time in a single sentence, perhaps word as "As well as being highly specific, this new molecular assay demonstrated amplification in under 15 minutes from AHG DNA". I would be more inclined to only focus on amplification time here, however, e.g. "This new molecular assay demonstrated amplification in under 15 minutes from AHG DNA."

Other suggested changes are:

Line 30 - suggest starting the sentence with "This included" rather than "Including"

Line 57 - remove the comma after "animals"

Line 65 - add a comma after "preservation"

Line 204 - remove the comma after "locus"

Line 477 - remove the comma after "scats"

Line 505 - this sentence was reworded in this version but needs clarification - the current wording is "molecular diagnostic approaches ... are unlikely to amplify from..." which doesn't make sense. Maybe "...inhibitors, with amplification unlikely in crude (scat) DNA extracts" or else "unlikely to be successful in..."

**Do you want your identity to be public for this peer review?** For information about this choice, including consent withdrawal, please see our Privacy Policy

Reviewer #2: No

Reviewer #3: **Yes:** Kathryn Wiltshire

---

## [Editor Report · Acceptance letter]

PONE-D-25-13902R3

PLOS One

Dear Dr. Blacket,

I'm pleased to inform you that your manuscript has been deemed suitable for publication in PLOS One. Congratulations! Your manuscript is now being handed over to our production team.

Kind regards,

on behalf of

Dr. Lei Zhang

Academic Editor

PLOS One